# IndiSeek learns information-guided disentangled representations

## Abstract

Learning disentangled representations is a fundamental task in multi-modal learning. In modern applications such as single-cell multi-omics, both shared and modality-specific features are critical for characterizing cell states and supporting downstream analyses. Ideally, modality-specific features should be independent of shared ones while also capturing all complementary information within each modality. This tradeoff is naturally expressed through information-theoretic criteria, but mutual-information-based objectives are difficult to estimate reliably, and their variational surrogates often underperform in practice. In this paper, we introduce IndiSeek, a novel disentangled representation learning approach that addresses this challenge by combining an independence-enforcing objective with a computationally efficient reconstruction loss that bounds conditional mutual information. This formulation explicitly balances independence and completeness, enabling principled extraction of modality-specific features. We demonstrate the effectiveness of IndiSeek on synthetic simulations, a CITE-seq dataset and multiple real-world multi-modal benchmarks.

## 1 Introduction

The growing availability of multi-modal data has broadened the scope of representation learning. One notable example is the development of single-cell multi-omics technologies in genomics (Teichmann & Efremova, 2020). For instance, CITE-seq (Cellular Indexing of Transcriptomes and Epitopes by Sequencing) (Stoeckius et al., 2017) enables simultaneous measurement of gene expression via single-cell RNA sequencing and protein abundance via antibody-derived tags (ADTs). Integrative analysis of both modalities at the single-cell resolution has improved tasks such as cell state delineation and enabled cell atlas construction and querying at finer granularity. Multi-modal data is also abundant at the intersection of computer vision, natural language processing, and audio processing, where combining information from different sources for representation learning forms the backbone of many groundbreaking progresses (Radford et al., 2021; Baltrušaitis et al., 2018; Jia et al., 2021; Akbari et al., 2021; Liang et al., 2024).

The central question of multi-modal learning is how to extract informative and interpretable representations from complex multi-modal data. This involves two complementary goals:

(1) Extracting interdependence (or *shared* information) across modalities;

(2) Extracting *modality-specific* information unique to each modality.

Shared features enhance predictive power across modalities by capturing cross-modal dependence. From an information-theoretic perspective, they should contain all information common across modalities. At the same time, preserving modality-specific diversity is equally important. In single-cell multi-omics, modality-specific features often correspond to unique biological signals that aid cell type identification and the discovery of novel or rare cell populations (Hao et al., 2021; Caron et al., 2025). After all, if all the information in a certain modality can be inferred from other modalities, there is little reason to measure it in the first place. Finally, disentangling shared and modality-specific components improves interpretability: shared features summarize cross-modal information, while modality-specific features capture complementary signals.

Methodologically, shared information is usually extracted by maximizing cross-modal mutual information (Tosh et al., 2021; Sridharan & Kakade, 2008), with contrastive methods like CLIP (Radford et al., 2021) widely adopted in practice. Under certain conditions, CLIP-learned representations have been shown to be not only sufficient (i.e., capturing all shared information), but also minimally sufficient (i.e., containing minimal extra information) (Gui et al., 2025; Oko et al., 2025; Lin & Mei, 2025) under certain conditions.

In contrast, identifying modality-specific features, also known as disentangled representation learning, has become an active research area (Fischer, 2020; Liang et al., 2023a; Liu et al., 2023; Wang et al., 2024a; Dufumier et al., 2024; Wang et al., 2024b). While shared features should be minimally sufficient, modality-specific features must both remain *independent* of (i.e., *disentangled* from) shared features and capture all *complementary* information within each modality (Liang et al., 2023a; Wang et al., 2024a;b). From an information-theoretic perspective, this requires (1) simultaneously minimizing the mutual information between shared and modality-specific features and (2) maximizing the mutual information between data and their union. Multiple state-of-the-art methods (Liang et al., 2023a; Wang et al., 2024a) tackle this problem from such a perspective, but differ in how they approximate the underlying mutual information. Since such quantities are difficult to estimate with finite samples, these methods often trade off independence against completeness. As we show in the motivating examples below, this tradeoff could interfere with their ability to achieve both goals simultaneously.

## 1.1 MOTIVATING EXAMPLES AND LIMITATIONS OF SOTA

To demonstrate the limitation of the state-of-the-art (SOTA) methods in disentangled learning, we consider two simulated settings. To isolate the challenge of modality-specific feature extraction, we assume the shared features $C_1 = f_1(X_1) \in \mathbb{R}^{d_c}$ are provided by an oracle. The task is then to learn modality-specific features from a single modality $X_1 \in \mathbb{R}^{d_1}$, with $d_1 = 6$ and $d_c = 2$.

- Setting 1: Let the observed data be iid copies of $X_1 \sim \mathcal{N}(0, I_{d_1})$. For each $x \in \mathbb{R}^{d_1}$, define the shared representation map

$$f_1(x) = 0.5 A_f x + 0.2 \sin(A_f x) + 0.2 (A_f x)^3 \quad \text{with} \quad A_f = (\mathbf{I}_{d_c}, \mathbf{O}) \in \mathbb{R}^{d_c \times d_1}.$$

Here, the sine and cubic functions are applied entrywise. The ideal modality-specific features are the last four coordinates of $X_1$, which contain all remaining information while being independent of $C_1$.

- Setting 2: Let the observed data be iid copies of $X_1$ where $(X_1)_{\{1,2,5,6\}} \sim \mathcal{N}(0, \mathbf{I}_4)$, $(X_1)_3 = 0.2 \times ((X_1)_1 + (X_1)_2)$, and $(X_1)_4 = (X_1)_1 \times (X_1)_2$. Thus, the third and fourth coordinates are deterministic functions of the first two, while the others are independent. Let $C_1 = (X_1)_{1:d_c}$. Here, the ideal modality-specific features are the last two coordinates.

For both settings, given $C_1$, our goal is to learn a map $h_1 : \mathbb{R}^{d_1} \to \mathbb{R}^{p_1}$ with $p_1 = 10$ such that $h_1(X_1)$ gives the modality-specific features. All neural networks are five-layer ReLU MLPs with width 100, trained on 10000 samples. Ablation studies beyond Gaussian distributions are presented in the appendix.

For any learned map $h_1$, we quantify the importance of each coordinate of $X_1$ in determining its output via gradients based on feature masking[1] on an independent test set of size 1000, following standard model-free importance measures (Robnik-Šikonja & Kononenko, 2008; Zeiler & Fergus, 2014; Li et al., 2016)

We present the average coordinate importance metrics in learned modality-specific representation maps over 50 simulation runs in Figures 1 and 2 for three different learning algorithms in the two settings, respectively. In both figures, the left panels report results of the SOTA approach FactorizedCL in Liang et al. (2023a) (without self-supervision), the middle panels report results of the SOTA approach InfoDisen in Wang et al. (2024a), and the right panels report results of the new approach IndiSeek we propose in this paper. Tuning parameters[2] are set at 0.1 for InfoDisen and IndiSeek and at 1.0 by default for FactorizedCL.

---

[1] See Appendix A.1 for details on this feature importance metric.

[2] See Appendix A.2 for an ablation study on performances with different tuning parameter values.

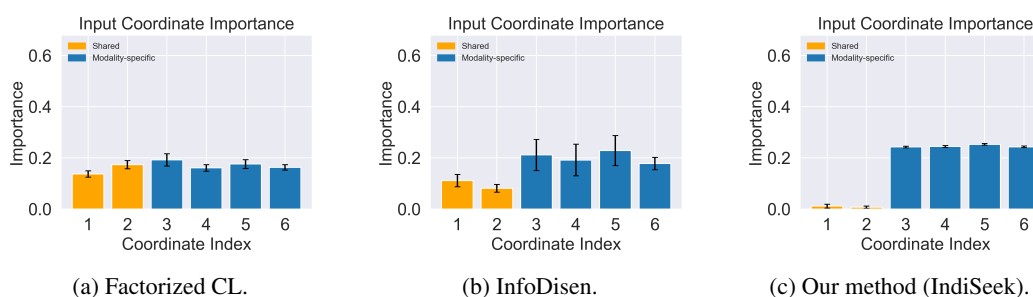

Figure 1: Importance of learned modality-specific features: Setting 1.

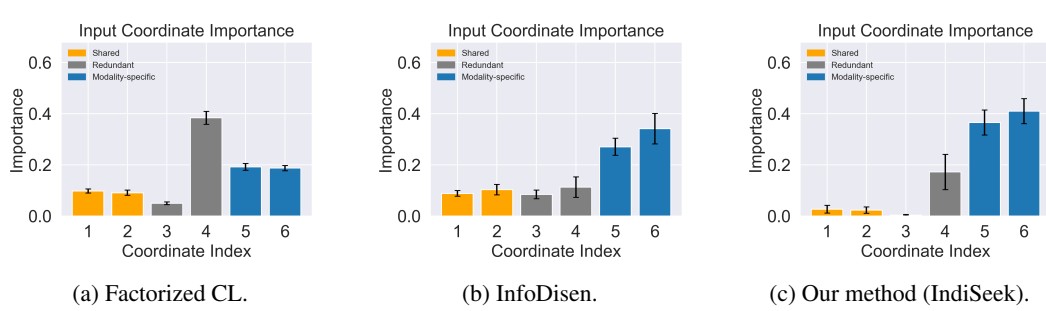

Figure 2: Importance of learned modality-specific features: Setting 2.

As we have reasoned when introducing the settings, the ideal modality-specific representations in Setting 1 should depend on and only on the last four coordinates, while in Setting 2, on and only the last two coordinates. The left and the middle panels of both figures suggest that neither FactorizedCL nor InfoDisen learn an ideal representation in either setting, as irrelevant coordinates play non-trivial roles in determining the learned modality-specific representations. Therefore, both SOTA approaches have their limitations in achieving the desired goal of disentangled representation learning. In contrast, the right panel of Figure 1 shows that the new method IndiSeek we are to propose achieves the ideal property: the learned modality-specific features depend on and only on all coordinates that are independent of those involved in shared features. In Figure 2, compared with SOTA methods, modality-specific representations learned by IndiSeek exhibit much less dependence on both shared and redundant features, which demonstrates its effectiveness in this setting with highly nonlinear dependence and nontrivial redundancy.

## 1.2 PAPER ORGANIZATION

In view of the foregoing limitations of SOTA approaches, in this paper, we introduce a new computationally efficient method IndiSeek, for extracting both shared and modality-specific features, and with a focus on the latter. Based on a reconstruction loss serving as an upper bound for mutual information, IndiSeek is capable of disentangling shared and modality-specific features, while avoiding information loss in each modality at the same time. The effectiveness of IndiSeek is demonstrated on both simulated and real-world datasets, including a single-cell multi-omics dataset CITE-seq, as well as multi-modal benchmark datasets.

The rest of this paper is organized as follows. Section 2 reviews the properties an ideal "shared + modality-specific" feature representation of multi-modal data should have. Section 3 introduces IndiSeek in detail as a practical implementation for seeking such an ideal representation and compares it with SOTA methods in terms of different choices made when approximating information-theoretic quantities. Experiments on a CITE-seq dataset and on MultiBench datasets are presented in Sections 4 and 5, respectively. Section 6 discusses other related works and potential extensions of the proposed method. Technical details and additional numerical studies are deferred to Appendices.

## 2 IDEAL DECOMPOSITION OF MULTI-MODAL DATA

We consider a *fully unsupervised* setting with observations from two modalities $X_1 \in \mathbb{R}^{d_1}$ and $X_2 \in \mathbb{R}^{d_2}$. Our goal is to decompose each modality into a shared information component that captures cross-modal information and a modality-specific information component that captures unique variation:

$$X_1 \mapsto (\underbrace{C_1}_{\text{shared}}, \underbrace{Z_1}_{\text{specific}}), \qquad X_2 \mapsto (\underbrace{C_2}_{\text{shared}}, \underbrace{Z_2}_{\text{specific}}).$$

Here $C_1$ and $C_2$ are $d_c$-dimensional[3] shared information components (though not necessarily identical) while $Z_1 \in \mathbb{R}^{p_1}$ and $Z_2 \in \mathbb{R}^{p_2}$ are modality-specific information components.

We formalize the ideal representation / decomposition by requiring the following desiderata:

1. **Minimal sufficiency:** the shared features preserve all information common to both modalities, i.e.,
$$I(X_1; X_2) = I(C_1; C_2).$$

   Intuitively, $C_1$ and $C_2$ should be sufficient for cross-modal prediction tasks. In addition, the shared features should not encode redundant details,
$$I(C_1; X_1) = \min\{I(f(X_1); X_1) : I(X_1; X_2) = I(f(X_1); X_2)\},$$

   and an analogous identity holds for $C_2$. This prevents "over-capturing" modality-specific signals inside the shared space.

2. **Independence:** modality-specific and shared features should be disentangled[4],
$$I(Z_1; C_2) = I(Z_2; C_1) = 0.$$

3. **Complementary information capture:** the pair $(C_i, Z_i)$ should retain enough information to recover the $i$th modality:
$$I(C_i, Z_i; X_i) \text{ is maximized for } i = 1, 2.$$

   This ensures that information not explained by shared features is fully captured by the modality-specific components.

Together, these criteria define what it means for a decomposition to be both *sufficient* and *disentangled*.

In practice, the extraction of shared features is often carried out with contrastive learning methods such as CLIP, which we also adopt in our framework. Therefore, in the remainder of this work, we focus primarily on the more challenging problem of learning *modality-specific* representations, assuming that the shared representations have already been obtained.

## 3 INFORMATION-GUIDED DISENTANGLED REPRESENTATION SEEKING (INDISEEK)

Given learned shared features, the desiderata in Section 2 motivate the search for disentangled modality-specific features, but leave open the question of *how to realize them*. Prior work has proposed an information-theoretic objective:

$$\max_{Z_i} \ I(Z_i, C_j; X_i) \qquad \text{subject to} \quad I(Z_i; C_{j'}) \approx 0, \tag{1}$$

---

[3]In theory, the dimensions of $C_1$ and $C_2$ need not match. In practice, these shared latent factors are often obtained from mapping both modalities into some co-embedding space. Therefore, without loss of generality, we assume that $C_1$ and $C_2$ have the same dimension.

[4]As $C_1$ is restricted to be a function of $X_1$, it is possible that $C_1$ also contains modality-specific information. Meanwhile, all information in $C_2$ about $X_1$ is strictly shared. Therefore, we require $I(Z_1; C_2) = I(Z_2; C_1) = 0$ instead of $I(Z_1; C_1) = I(Z_2; C_2) = 0$.

Figure 3: IndiSeek: Information-guided Disentangled Representation Seeking.

where $j, j' \in \{1, 2\}$ have respective specifications in each method, in which cross-modal switching of shared features is adopted when $j, j' \neq i$. This objective reflects the prevailing philosophy: modality-specific features should be maximally informative about their own modality, conditional on the shared component, while being independent of it.

Our position is not only to refine this conceptual objective to relieve the issue with redundancy in shared features, but also to propose an effective and tractable way to implement it in practice. In the following subsections, we describe a two-stage strategy to this end: (1) learning shared features via CLIP, and (2) extracting modality-specific features with a reconstruction-guided objective conditional on learned shared features in step (1). See Figure 3 for an overview of IndiSeek.

### 3.1 STEP 1: EXTRACTING SHARED FEATURES VIA CLIP

As noted at the end of Section 2, shared feature extraction is commonly handled by contrastive learning methods such as CLIP (Radford et al., 2021), which we also adopt. Concretely, encoders $(f_1, f_2)$ are trained with the InfoNCE loss (Oord et al., 2018), which provides a lower bound on the mutual information between $X_1$ and $X_2$. This encourages $C_1 = f_1(X_1)$ and $C_2 = f_2(X_2)$ to capture all cross-modal dependence while avoiding modality-specific redundancy.

### 3.2 STEP 2: EXTRACTING MODALITY-SPECIFIC FEATURES

Given the shared features $(C_1, C_2)$, we aim to extract the modality-specific components $(Z_1, Z_2)$ by operationalizing the constrained problem in Eq. (1). Here, we note that, if the learned shared representations are redundant, that is, $C_1$ contains more information than the underlying shared features, learning the unique feature $Z_1$ by encouraging the independence between $Z_1$ and $C_1$ is lossy: the learned $Z_1$ may lose modality-specific information. To address this, we propose the *cross-modal disentanglement*:

$$\min_{Z_i} \quad \underbrace{I(Z_i; C_{3-i})}_{\text{disentanglement-enforcing term}} \quad - \quad \lambda \cdot \underbrace{I(Z_i, C_i; X_i)}_{\text{complementary information capture term}} \quad .$$

The key is in the disentanglement term $I(Z_1, C_2)$ (or $I(Z_2, C_1)$). Even when the learned "shared" information is redundant, i.e., when $C_1$ and $C_2$ both contain the underlying shared features (say, $C^*$), the redundancy is only a function of modality-specific features in $X_2$ that is independent of $X_1$; thus, given that $I(C^*; Z_1) = I(C_2; Z_1)$, we ought to encourage the independence between $Z_1$ and $C_2$ instead.

Here, the minimization over $Z_i$ is in fact minimizing over functions $h_i(\cdot)$ on $X_i$. This form highlights the tradeoff: reducing the dependence of $Z_i$ on the shared representation while maximizing the collective informativeness of $(Z_i, C_i)$ about $X_i$. However, both terms involve mutual information, which is intractable to optimize directly.

To make this objective implementable, we replace each term with a bound that aligns with its optimization direction:

- For the disentanglement enforcing term $I(Z_i; C_{3-i})$, which we aim to minimize, we replace it by an **upper bound** given by the NCE-CLUB loss (Cheng et al., 2020; Liang et al., 2023a):

$$\mathcal{L}_{\text{NCE-CLUB}}(Z_1; C_2) = \mathbb{E}_{(Z_1, C_2)}\left[\log p(Z_1 \mid C_2)\right] - \mathbb{E}_{Z_1, \widetilde{C}_2}\left[\log p(Z_1 \mid \widetilde{C}_2)\right],$$

where $p(z_1 \mid c_2)$ is the underlying conditional density of $Z_1 \mid C_2$ and $\widetilde{C}_2$ is an independent copy of $C_2$ from the same marginal distribution.[5]

- For the complementary information capture term $I(Z_i, C_i; X_i)$, which we aim to maximize, we replace it by a **lower bound** realized through a reconstruction loss that serves as a surrogate for conditional mutual information.

This yields the practical IndiSeek objective that we aim to minimize:

$$\mathcal{L}_{\text{IndiSeek}}(Z_i; C_1, C_2, \lambda) \;=\; \mathcal{L}_{\text{NCE-CLUB}}(Z_i; C_{3-i}) \;+\; \frac{\lambda}{2\mathbb{E}\|X_i\|^2} \min_{g_i} \mathbb{E}\|g_i(Z_i, C_i) - X_i\|^2. \quad (2)$$

This formulation encourages $Z_i$ to be independent of the shared features while still retaining sufficient information measured via the optimal reconstruction error. Because the objectives are separable across modalities, the extraction procedure can be parallelized.

**Justification via reconstruction as MI bound.** When $C_1$ is fixed, maximizing $I(Z_1, C_1; X_1) = I(Z_1; X_1 \mid C_1) + I(X_1; C_1)$ is equivalent to maximizing $I(Z_1; X_1 \mid C_1) = H(X_1 \mid C_1) - H(X_1 \mid Z_1, C_1)$, which is further equivalent to minimizing the conditional entropy $H(X_1 \mid Z_1, C_1)$. By Fano's inequality (Cover, 1999, Theorem 8.6.6), the reconstruction error upper bounds this conditional entropy for any $g_1$:

$$H(X_1 \mid Z_1, C_1) \;\leq\; \tfrac{1}{2} \log\Big(2\pi e \cdot \mathbb{E}\|X_1 - g_1(Z_1, C_1)\|^2\Big),$$

where equality holds if and only if the conditional distribution of $X_1 \mid (Z_1, C_1)$ is Gaussian. This shows why quadratic reconstruction loss is a principled surrogate for conditional mutual information.

### 3.3 Comparison with two SOTA Methods

Recent approaches to disentangled multi-modal representation learning differ mainly in how they approximate the two terms in Eq. (1): the disentanglement enforcing term $I(Z_i, C_j)$, and the complementary information capture term $I(Z_i, C_{j'}; X_i)$, where $j, j' \in \{1, 2\}$ will be specified in each method.

**Factorized Contrastive Learning (Factorized CL, Liang et al., 2023a).** Like IndiSeek, this method uses the NCE-CLUB loss to upper bound the entanglement enforcing term. However, it replaces the complementary information capture term with an InfoNCE lower bound. For comparison, we consider its task-agnostic, two-step variant:

$$\mathcal{L}_{\text{FactorizedCL}}(Z_1; C_1, C_2, \tau, \lambda) = \mathcal{L}_{\text{NCE-CLUB}}(Z_1; C_1) + \frac{\lambda}{2}\mathcal{L}_{\text{infoNCE}}(h(C_1, Z_1), X_1, \tau).$$

We note that here $j, j'$ in (1) both equal $i$, i.e., no cross-modal switching is implemented. Liang et al. (2021) also proposes a self-supervised variant of Factorized CL by leveraging task-relevant augmentations, more discussions on which are deferred to Section 6 and Appendix A.1. In this paper, we adapt the unsupervised version of Factorized CL, i.e., we use `CLUBInfoNCECritic` between corresponding features in Liang et al. (2023b) as the disentanglement objective, and adopt the implementation of CLUB loss in Cheng et al. (2020) in both IndiSeek and Factorized CL for fair comparison. (more details are presented in Appendix A.1).

**Disentangled Self-Supervised Learning (InfoDisen Wang et al., 2024a).** This method also relies on InfoNCE for the complementary information capture term, but replaces NCE-CLUB with an orthogonal loss based on von Mises–Fisher assumptions:

$$\mathcal{L}_{\text{InfoDisen}}(Z_1; C_1, C_2, \tau, \lambda) = \mathbb{E}_{(Z_1, C_1)} \left[\langle \mu(Z_1), \mu(C_1)\rangle\right] + \frac{\lambda}{2}\mathcal{L}_{\text{infoNCE}}(h(C_2, Z_1), X_1, \tau).$$

We note that, in the reconstruction term, Wang et al. (2024a) adopts the shared feature from the other modality, i.e., the actual reconstruction of interest becomes $I(Z_i, C_{3-i}; X_i)$ instead of $I(Z_i, C_i; X_i)$ by replacing $j$ with $3 - i$ in (1).

---

[5]In implementation, following Cheng et al. (2020), we replace conditional distributions by their estimates via multivariate Gaussian distributions whose moments are fitted from data in each epoch.

Both methods inherit limitations: InfoNCE is sensitive to temperature tuning $\tau$ and may underperform as a surrogate for capturing complementary information, while the orthogonal loss assumes linearity and fails to capture nonlinear dependencies. Additionally, without cross-modal disentanglement, InfoDisen and FactorizedCL may fail to fully capture modality-specific features due to redundant shared features. IndiSeek avoids these issues by pairing NCE-CLUB with a reconstruction-based mutual information bound.

## 4 EXPERIMENTS WITH A CITE-SEQ DATASET

In this section, we evaluate the performance of IndiSeek on applications in single-cell biology. We consider the bone marrow CITE-seq dataset from Stuart et al. (2019) which consists of measurements in two modalities on 30672 individual cells: transcriptome (RNA) and 25 cell-surface proteins (ADT). We randomly split the dataset into a training set of size 15000 and a test set of size 15672. To leverage both RNA and ADT data for annotating cell types, Hao et al. (2021) proposed to cluster cells according to a weighted-nearest-neighbor (WNN) graph with cells as nodes. In its construction, the similarity between a cell $i$ and every other cell $j$ is measured by a weighted average of their respective similarities in the two modalities. The weights used by cell $i$ are in turn determined by the cross-modal predictive powers of its individual modalities after local smoothing. Thus, the RNA weight of each cell can be viewed as a quantification of the importance RNA plays in determining its cell state. The larger, the more important. Based on clustering nodes of the WNN graph, Hao et al. (2021) annotated cells at two granularity levels with 5 (level-1) and 27 (level-2) cell types, respectively.

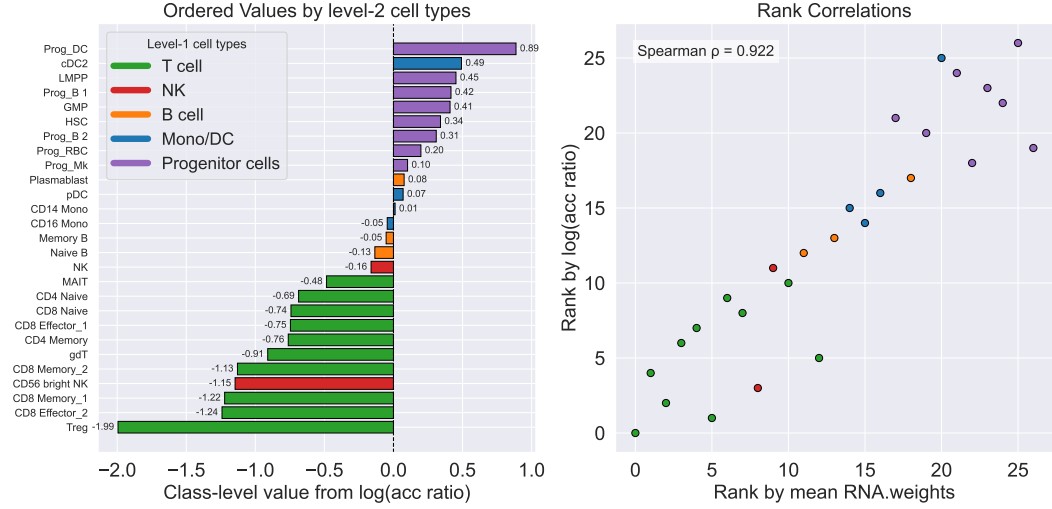

Figure 4: Performance of IndiSeek in CITE-seq dataset ($\lambda = 10.0$).

In this experiment, we first apply IndiSeek to train neural nets[6] that map RNA and ADT data to their disentangled representations, $(C_{\text{RNA}}, C_{\text{ADT}})$ and $(Z_{\text{RNA}}, Z_{\text{ADT}})$, using the training set. We then apply the trained networks to infer disentangled representations of cells in the test set. The ideal disentangled representations should reflect distinctive levels of informativeness that individual modalities have in determining each cell's annotation. To this end, for each cell $c$, we find its 10 nearest neighbors measured by Euclidean distance in $Z_{\text{RNA}}$ and $Z_{\text{ADT}}$, respectively, and compare the proportions of neighbors with the same level-2 annotations as $i$ in the two modalities, denoted by $\beta_{\text{RNA}}(c)$ and $\beta_{\text{ADT}}(c)$, respectively. If the two proportions differ sizably, the modality-specific information in the higher-proportion modality plays a more important role in determining the cell's annotation, which motivates us to summarize the comparison with a score $\theta(c) = \log(\beta_{\text{RNA}}(c)/\beta_{\text{ADT}}(c))$ that is monotone increasing with respect to the importance of RNA-specific information.

---

[6]In this example, representation maps are trained within the class of 5-layer ReLU neural networks with middle layers of width 50.

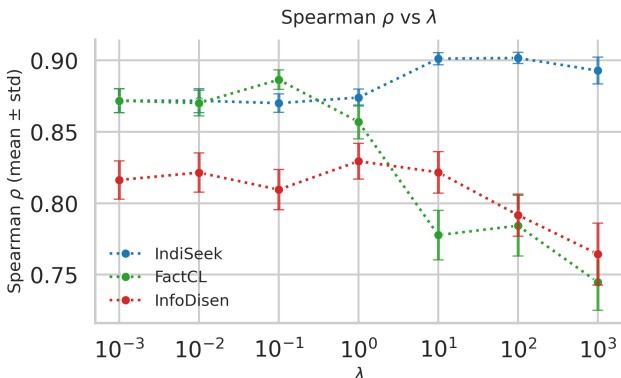

Figure 5: Comparison of rank correlation metrics across three methods on the CITE-seq dataset.

In the left panel of Figure 4, 27 level-2 cell types are ranked according to average $\theta(c)$ scores of cells in the test set with respective cell type annotations, in which a higher rank indicates a higher impact of the RNA-specific information in determining the cell type. Each bar is colored according to the coarser level-1 cell type that each level-2 cell type belongs to for easier visual inspection. Different cell types within the T cell family have negative scores, indicating RNA-specific information is less informative compared to ADT-specific counterpart for differentiating different T cell subpopulations, which is in line with the observation that clustering transcriptomics data typically does not delineate T cell subpopulations (Szabo et al., 2019; Zheng et al., 2021) and gating based on surface protein markers is often needed (Kotliar et al., 2025). In contrast, RNA-specific information plays a more important role in distinguishing different progenitor cells than ADT-specific information. To further demonstrate the quality of learned modality-specific representations, we plot the ranks of average $\theta(c)$ values against the benchmark ranks of average RNA weights in Hao et al. (2021) of the 27 level-2 cell types in the right panel in Figure 4, which exhibit a strong correlation that is confirmed by Spearman's rank correlation (Spearman, 1961) of 0.910. In Appendix A.3, we present UMAPs of IndiSeek-learned representations in Figures 18 and 21, which exhibit clear clusters with respect to two levels of cell types that are also confirmed with metrics such as ARI and NMI (details are given in Appendix A.3). As a comparison, we repeated the foregoing experiments across 10 random seeds while replacing IndiSeek with InfoDisen or FactorizedCL when training the same neural net architecture and the resulting alignment metrics with respect to the results in Hao et al. (2021) are presented in Figure 5, in which we also report the error bars for results over 10 random seeds. Both alternative methods yielded inferior alignment with the relative importance of the RNA modality in Hao et al. (2021) compared to IndiSeek. To ablate the choice of $\lambda$, in all experiments with real-world data, we report results with varying $\lambda$ on grid values $10^j$ for $j = -3, -2, -1, 0, 1, 2, 3$. Additionally, UMAPs of representations learned by InfoDisen and FactorizedCL are also presented in Appendix A.3 Figures 19, 20, 22, and 23.

## 5 EXPERIMENTS WITH MULTIBENCH DATASETS

In addition to the application in single-cell multi-omics, we also evaluate the performance of IndiSeek on MultiBench datasets (Liang et al., 2021), which include 4 video datasets (MOSI (Zadeh et al., 2016), MOSEI (Zadeh et al., 2018), UR-FUNNY (Hasan et al., 2019), MUStARD (Castro et al., 2019)) and Medical Information Mart for Intensive Care III (MIMIC) dataset (Johnson et al., 2016). We follow the processing pipeline in Liang et al. (2023a) and adopt exactly the same data splitting and feature pre-extraction, in which we focus on the task of binary classification on whether the patient fits any ICD-9 code in group 7 for the MIMIC dataset. We use a smaller Transformer architecture with 2 heads, 2 layers, and the intermediate layers with a width 128. For all methods in comparison, we train the architecture for 2000 epochs with the same training parameters. We note that our focus here is to benchmark the qualities of representations learned with task-agnostic objective functions. Therefore, for fair comparison along this direction, we do not involve task-related within- or cross-modality data augmentations and hence do not involve self-supervised learning terms in objective functions of all methods in comparison. In contrast, to achieve the optimal perfor-

mance in a specific task on a particular benchmark dataset, appropriate data augmentation informed by the task and domain knowledge is often helpful.

Following Liang et al. (2023a) and Wang et al. (2024a), we focus on the linear probing accuracy for each method with learned representations $(C_1, Z_1, C_2, Z_2)$ as features, i.e., we concatenate shared and modality-specific features for classification. Here we also compare with the CLIP baseline, where only the shared representations $(C_1, C_2)$ are used for linear probing accuracy evaluation. Following the same training procedure as for the CITE-seq dataset, we report the results with three methods in Table 1. For each method, we vary $\lambda$ in $\{10^j : j = -3, -2, -1, 0, 1, 2, 3\}$ and for each $\lambda$, we run each method with 10 random seeds. Table 1 presents the averaged accuracy for the best choice of $\lambda$. More details on the performance with varying $\lambda$'s can be found in Appendix A.4.1.

From Table 1, we can see that IndiSeek outperms baseline and two SOTA methods across all datasets, and the performances of three disentangled learning methods are comparable in datasets MUStARD and MIMIC, among which, MUStARD (690 samples) has a small scale. Moreover, all three disentangled learning methods outperform the CLIP baseline in most of the datasets, which demonstrates the necessity of involving modality-specific features in downstream tasks. In video datasets, such as MOSI (2199 samples), MOSEI (22777 samples), and UR-FUNNY (16514 samples), IndiSeek exhibits more pronounced performance gain in accuracy under the same training configurations, indicating its efficiency in enforcing disentanglement and preserving sufficiency in representations.

**Ablation studies with varying output dimension.** To investigate the robustness of IndiSeek across different output dimension capacities, we conduct experiments on the MOSI dataset with output dimensions $d \in \{20, 60, 100, 140, 180\}$. For each output dimension, we perform a sweep over $\lambda$ and report the maximum accuracy achieved across all $\lambda$ values, averaged over 10 random seeds. Table 2 presents the results of this analysis. IndiSeek consistently achieves the best performance across all output dimensions, demonstrating its robustness to this hyperparameter choice.

Table 1: Comparison of accuracy on multimodal datasets (averaged over 10 seeds, standard errors in parentheses, max average over $\lambda$). All values are percentages.

| Method | MOSI | MOSEI | UR-FUNNY | MUStARD | MIMIC |
|---|---|---|---|---|---|
| IndiSeek | $\mathbf{70.03}_{(1.39)}$ | $\mathbf{75.47}_{(0.13)}$ | $\mathbf{63.79}_{(0.39)}$ | $\mathbf{57.46}_{(1.04)}$ | $\mathbf{65.99}_{(0.31)}$ |
| FactorizedCL | $67.11_{(0.34)}$ | $74.74_{(0.04)}$ | $58.36_{(0.25)}$ | $56.45_{(1.16)}$ | $65.69_{(0.11)}$ |
| InfoDisen | $67.52_{(0.62)}$ | $74.73_{(0.08)}$ | $58.08_{(0.50)}$ | $56.16_{(0.92)}$ | $65.47_{(0.23)}$ |
| CLIP (baseline) | $67.61_{(0.66)}$ | $74.70_{(0.05)}$ | $58.32_{(0.68)}$ | $55.36_{(1.12)}$ | $64.56_{(0.29)}$ |

Table 2: MOSI results with varying output dimensions (averaged over 10 seeds, standard errors in parentheses, max average over $\lambda$). All values are percentages.

| Method | outdim=20 | outdim=60 | outdim=100 | outdim=140 | outdim=180 |
|---|---|---|---|---|---|
| IndiSeek | $\mathbf{69.68}_{(0.51)}$ | $\mathbf{70.03}_{(0.80)}$ | $\mathbf{69.56}_{(0.57)}$ | $\mathbf{70.20}_{(0.65)}$ | $\mathbf{68.80}_{(1.15)}$ |
| FactorizedCL | $69.49_{(0.26)}$ | $66.91_{(1.04)}$ | $67.52_{(0.95)}$ | $68.28_{(0.67)}$ | $67.08_{(0.29)}$ |
| InfoDisen | $68.57_{(0.91)}$ | $65.95_{(1.59)}$ | $65.17_{(0.68)}$ | $66.90_{(0.33)}$ | $66.62_{(0.68)}$ |
| CLIP (baseline) | $69.46_{(0.45)}$ | $67.03_{(0.79)}$ | $66.53_{(0.68)}$ | $68.54_{(0.94)}$ | $67.20_{(1.04)}$ |

**Multi-task performance.** To further evaluate the learned task-agnostic representations, we also investigate the multi-task performance of each method on the MIMIC dataset. We consider the tasks of predicting ICD-9 codes for multiple groups in the MIMIC dataset in MultiBench (Liang et al., 2021). We compare the performance of all three disentangled learning methods together with CLIP baseline in Table 3, in which we calculate the average accuracy across tasks of predicting the last three ICD-9 codes, and present results in a subset of ICD-9 codes. Complete results for other tasks are deferred to the appendix.

Table 3: Multi-task performance (averaged over 10 seeds, standard errors in parentheses, max average over $\lambda$) on MIMIC dataset (ICD-9 groups 17–19). All values are percentages.

| Method | group 17 | group 18 | group 19 | Average (3 tasks) |
|---|---|---|---|---|
| IndiSeek | $\mathbf{62.58}_{(0.17)}$ | $\mathbf{60.99}_{(0.19)}$ | $\mathbf{69.71}_{(0.11)}$ | $\mathbf{64.43}$ |
| FactorizedCL | $61.83_{(0.20)}$ | $60.36_{(0.14)}$ | $69.30_{(0.12)}$ | $63.83$ |
| InfoDisen | $61.72_{(0.14)}$ | $60.40_{(0.34)}$ | $69.46_{(0.14)}$ | $63.86$ |
| CLIP (baseline) | $61.50_{(0.14)}$ | $60.71_{(0.09)}$ | $69.25_{(0.16)}$ | $63.82$ |

## 6 EXTENSIONS AND DISCUSSION

Guided by information-theoretic principles, we proposed IndiSeek, a method that extracts shared and modality-specific features that are simultaneously sufficient and disentangled. While our focus has been on task-agnostic disentangled representation learning, IndiSeek can be readily extended to task-related settings.

**Task-related extensions.**

- **With task-specific augmentations.** When a downstream task is known in advance, domain knowledge can be used to design within-modality data augmentations. If the augmentations are *ideal* in the sense that the task-invariant orbit of each observation is fully covered (Liang et al., 2023a), one may first perform within-modality contrastive learning. The resulting representations retain only task-relevant information. IndiSeek can then be applied to further decompose these task-relevant features into shared and modality-specific components.

- **With side information** $Y$**.** When auxiliary task-related information $Y$ is available, IndiSeek can treat $Y$ as an additional modality and be applied to $(X_i, Y)$ pairs. The modality-specific features of each $X_i$ relative to $Y$ can then guide the design of effective augmentations, after which the procedure above can be repeated on the task-related representations. In the extreme case where $Y$ is the task label itself, each $X_i$ can first be reduced to its shared features with $Y$, denoted $\widetilde{X}_i$, since $\widetilde{X}_i$ contains all information in $X_i$ about $Y$. IndiSeek may then be applied to $\widetilde{X}_1, \widetilde{X}_2$ for further decomposition.

**Tuning parameter selection.** The selection of the Lagrange multiplier $\lambda$ in (2) is important for the implementation of IndiSeek. With the rescaled reconstruction loss, both the CLUB upper bound and the reconstruction term are scale-free, and our experimental results have suggested a candidate range of $\lambda$'s between $10^{-1}$ to $10^{1}$. Identifying the precise optimal choice of $\lambda$ remains an interesting future direction to enhance empirical performance and theoretical understanding.

**Additional related work.** In the literature, there are other alternatives that learn both shared and modality-specific features in an information-theoretic manner. For example, Dufumier et al. (2024) adopts infoNCE as a proxy for mutual information. However, the framework in Dufumier et al. (2024) cannot ensure disentanglement between shared and modality-specific features, leading to potential redundancy in learned representations. In a concurrent work, Shi et al. (2025) adopts variational bounds for mutual information and entropy regularization as objectives to learn disentangled features. However, similar to Liang et al. (2023a) and Dufumier et al. (2024), the objectives are motivated by some task labels $Y$ and hence differ from our goal of learning task-agnostic features.

## REPRODUCIBILITY STATEMENT

Reproducible codes for numerical experiments on simulated datasets in Section 1.1, the CITE-seq dataset in Section 4, and MultiBench datasets in Section 5 are attached to the supplementary materials of this submission.

## LLM USAGE DISCLOSURE

We used large language models (LLMs) to refine the implementation code.

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

## A    ADDITIONAL EXPERIMENTAL RESULTS

### A.1    IMPLEMENTATION DETAILS AND METRICS

**Masking-based feature importance.**    In simulated experiments, to determine which coordinates are extracted as shared or modality-specific features in $X_1$, we consider model-free feature importance based on feature masking related to Robnik-Šikonja & Kononenko (2008); Zeiler & Fergus (2014); Li et al. (2016). Concretely, given a black-box architecture $\psi : \mathbb{R}^d \to \mathbb{R}^p$, to understand which coordinates in each $x \in \mathbb{R}^d$ are effective in the output $\psi(x)$, we use leave-one-coordinate-out $x_{-j}$ as inputs for $j \in [d]$ and quantify the difference between and original output and output with masked input $\|\psi(x) - \psi(x_{-j})\|^2$. In implementation, for each coordinate $j \in [d]$, we draw $M = 1000$ random vectors $\{x_i\}_{i \in [M]}$ from uniform distributions on $[-10, 10]^d$ and calculate the average

$$\zeta_\psi(j) = \frac{1}{M} \sum_{i \in [M]} \|\psi(x_i) - \psi(x_{i,-j})\|^2.$$

In this paper, we normalize the vector $\boldsymbol{\zeta}_\psi = (\zeta_\psi(j))_{j \in [p]}$ to the simplex with unit $\ell_1$-norm and use this normalized score $\widehat{\zeta}_\psi(j)$ to measure the importance of the $j$th coordinate in $\psi$.

**Rank correlation.**    Recall that we obtain disentangled representations for RNA and ADT data, $(C_{\text{RNA}}, C_{\text{ADT}})$ and $(Z_{\text{RNA}}, Z_{\text{ADT}})$, using the training set. To this end, for each cell $c$, we find its 10 nearest neighbors measured by Euclidean distance in $Z_{\text{RNA}}$ and $Z_{\text{ADT}}$, respectively, and compare the proportions of neighbors with the same level-2 annotations as $i$ in the two modalities, denoted by $\beta_{\text{RNA}}(c)$ and $\beta_{\text{ADT}}(c)$, respectively. We summarize the comparison with a score $\theta(c) = \log(\beta_{\text{RNA}}(c)/\beta_{\text{ADT}}(c))$ that is monotone increasing with respect to the importance of RNA-specific information, and rank 27 cell types based on this metric. We use the RNA weights based on weighted k-NN obtained in Hao et al. (2021) as the benchmark and compare our ranks $(R_i)_{i \in [L]}$ for cell types with the RNA-weights-based ranks $(R_i^*)_{i \in [L]}$ with $L = 27$. We adopt Spearman's $\rho$ correlation as the metric, where for any two vectors of ranks as permutations of $[L]$,

$$\rho(\mathbf{R}, \mathbf{R}^*) = \frac{\sum_{i \in [L]} \left(R_i - \frac{L(L+1)}{2}\right)\left(R_i^* - \frac{L(L+1)}{2}\right)}{\sqrt{\sum_{i \in [L]} \left(R_i - \frac{L(L+1)}{2}\right)^2} \sqrt{\sum_{i \in [L]} \left(R_i^* - \frac{L(L+1)}{2}\right)^2}}.$$

Moreover, when there are no ties in both $\mathbf{R}$ and $\mathbf{R}^*$, it holds that

$$\rho(\mathbf{R}, \mathbf{R}^*) = 1 - \frac{6 \sum_{i \in [L]} (R_i - R_i^*)^2}{L(L^2 - 1)}.$$

**Implementation of CLUB loss.**    We follow the implementation of CLUB in Cheng et al. (2020). As CLUB loss is a variational upper bound for mutual information, we use an auxiliary conditional density $q_\theta(z|c)$ as an approximation for $p(c|z)$, where $q_\theta(z|c)$ lies in the Gaussian family, and we optimize the moments of the Gaussian distribution using 5-layer MLPs in each epoch. The inner optimization is run for 5 epochs at each epoch to optimize representations, and the implementation the CLUB loss is the same for both IndiSeek and FactorizedCL for fair comparison.

Liang et al. (2021) also proposes a self-supervised variant of Factorized CL by leveraging task-relevant augmentations, more discussions on which can be found in Section 6. In this paper, we adapt the unsupervised version of Factorized CL, i.e., we use `CLUBInfoNCECritic` between corresponding features in Liang et al. (2023b) as the disentanglement objective, and adopt the implementation of CLUB loss in Cheng et al. (2020) in both IndiSeek  and Factorized CL for fair comparison.

### A.2    NUMERICAL SIMULATIONS

In this section, we present additional results for the same setting in the main paper with varying $\lambda$'s. Recall the setting with two modalities with $d_1 = 6$ and $d_c = 2$:

- Setting 1: Let the observed data be iid copies of $X_1 \sim \mathcal{N}(0, I_{d_1})$. For each $x \in \mathbb{R}^{d_1}$, define the shared representation map

$$f_1(x) = 0.5 A_f x + 0.2 \sin(A_f x) + 0.2 (A_f x)^3 \quad \text{with} \quad A_f = (\mathbf{I}_{d_c}, \mathbf{O}) \in \mathbb{R}^{d_c \times d_1}.$$

  Here, the sine and cubic functions are applied entrywise. The ideal modality-specific features are the last four coordinates of $X_1$, which contain all remaining information while being independent of $C_1$.

- Setting 2: Let the observed data be iid copies of $X_1$ where $(X_1)_{\{1,2,5,6\}} \sim \mathcal{N}(0, \mathbf{I}_4)$, $(X_1)_3 = 0.2 \times ((X_1)_1 + (X_1)_2)$, and $(X_1)_4 = (X_1)_1 \times (X_1)_2$. Thus, the third and fourth coordinates are deterministic functions of the first two, while the others are independent. Let $C_1 = (X_1)_{1:d_c}$. Here, the ideal modality-specific features are the last two coordinates.

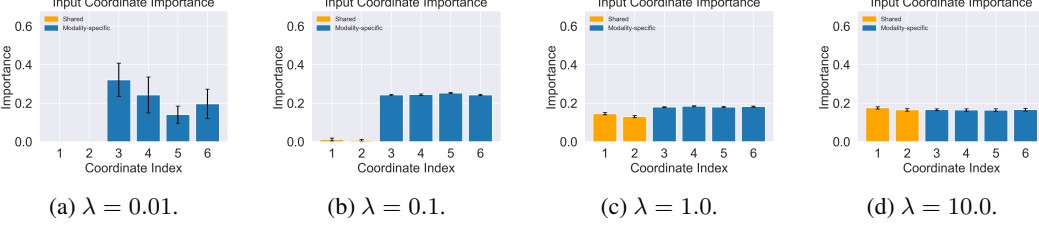

(a) $\lambda = 0.01$.  (b) $\lambda = 0.1$.  (c) $\lambda = 1.0$.  (d) $\lambda = 10.0$.

Figure 6: IndiSeek with varying $\lambda$: feature importance averaged over 50 simulations (Setting 1).

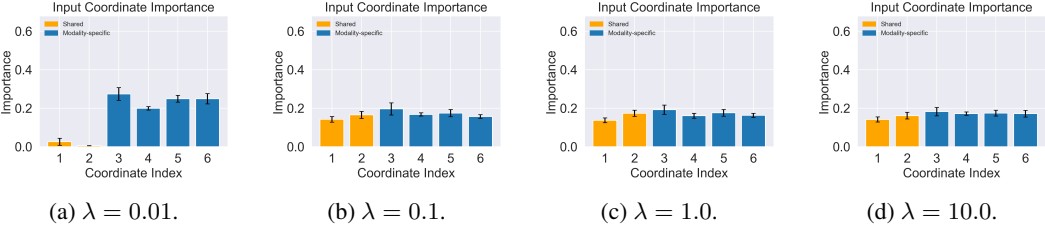

(a) $\lambda = 0.01$.  (b) $\lambda = 0.1$.  (c) $\lambda = 1.0$.  (d) $\lambda = 10.0$.

Figure 7: Factorized CL with varying $\lambda$: feature importance averaged over 50 simulations (Setting 1).

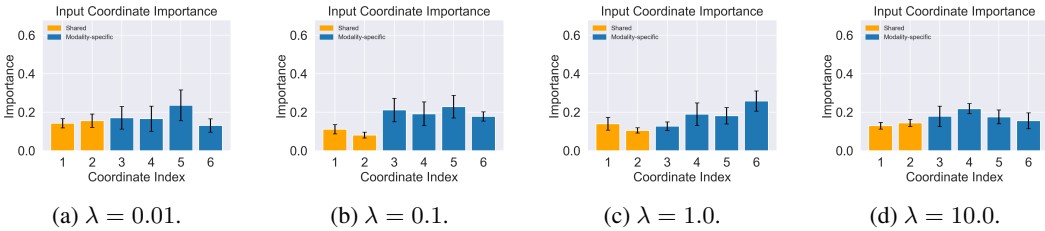

(a) $\lambda = 0.01$.  (b) $\lambda = 0.1$.  (c) $\lambda = 1.0$.  (d) $\lambda = 10.0$.

Figure 8: InfoDisen with varying $\lambda$: feature importance averaged over 50 simulations (Setting 1).

For Setting 1, given shared features are generated by sine and cubic functions, we can see from Figure 6 and 7 that both Factorized CL and IndiSeek are capable of extracting modality-specific features that are independent of shared ones when $\lambda = 0.01$, while IndiSeek tend to be more robust to the choice of $\lambda$ and preserves the desired performance when $\lambda = 0.1$. However, InfoDisen, due to the limitation of orthogonal loss in handling nonlinear dependence, fails to disentangle modality-specific features from shared ones regardless of the value of $\lambda$.

From Figures 9, 10, and 11, we can see that similar to the results presented in Section 1.1, with $\lambda$ in the range of $\{10^j : j = -2, -1, 0, 1\}$, InfoDisen fails to enforce independence between shared and modality-specific features; FactorizedCL tends to exclude shared features with a small $\lambda$ but still captures at least one of redundant features that is correlated with shared ones. In comparison, with $\lambda = 0.01$, IndiSeek can exactly capture the remaining two modality-specific features. Moreover,

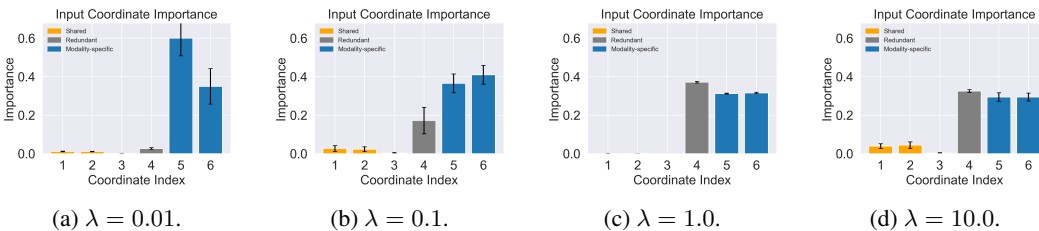

(a) $\lambda = 0.01$.     (b) $\lambda = 0.1$.     (c) $\lambda = 1.0$.     (d) $\lambda = 10.0$.

Figure 9: IndiSeek with varying $\lambda$: feature importance averaged over 50 simulations (Setting 2).

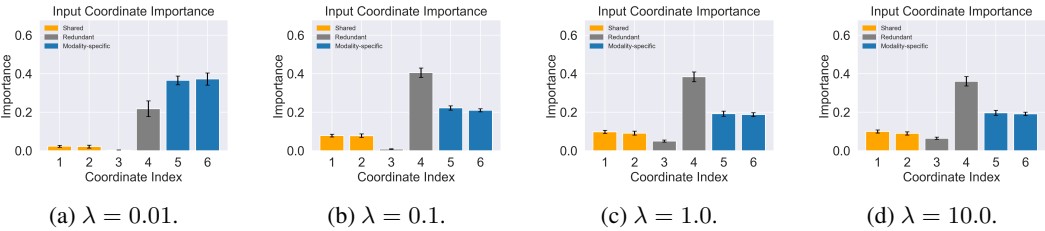

(a) $\lambda = 0.01$.     (b) $\lambda = 0.1$.     (c) $\lambda = 1.0$.     (d) $\lambda = 10.0$.

Figure 10: Factorized CL with varying $\lambda$: feature importance averaged over 50 simulations (Setting 2).

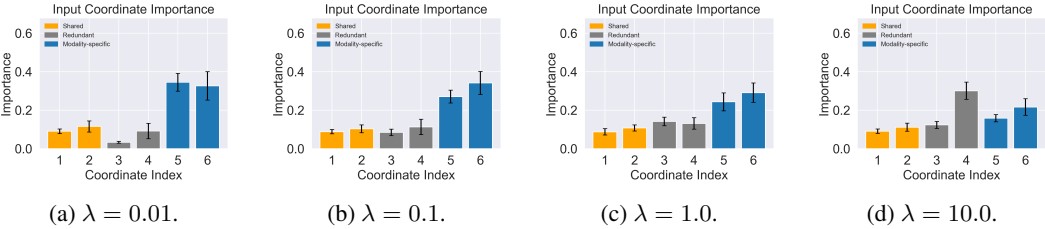

(a) $\lambda = 0.01$.     (b) $\lambda = 0.1$.     (c) $\lambda = 1.0$.     (d) $\lambda = 10.0$.

Figure 11: InfoDisen with varying $\lambda$: feature importance averaged over 50 simulations (Setting 2).

these figures also reveal the tradeoff between the disentanglement and sufficiency, determined by the value of $\lambda$. With $\lambda = 0.01$, more weight is put on the disentanglement between shared and modality-specific features; thus, with a strong regularization on dependence, IndiSeek tends to exclude all shared and redundant features. On the other hand, with $\lambda = 10$, the loss puts more weight on reconstruction/sufficiency, and as a result, disentangled methods may extract features that are dependent on the shared ones.

### A.2.1 ABLATION STUDIES BEYOND GAUSSIANITY

We also consider ablated variants of Settings 1 and 2 in Section 1.1. Concretely, we generate modality-specific input coordinates (i.e., $(X_1)_{d_c+1:d}$ in Setting 1 and $(X_1)_{d_c+3:d}$ in Setting 2) independently from a discrete distribution

$$(X_1)_j \sim \text{Unif}\{-\sqrt{7}/2, -1, -1/2, 1/2, 1, \sqrt{7}/2\}.$$

Results for variants (Setting 3 as a variant for Setting 1, and Setting 4 as a variant for Setting 2) are presented in Figures 12–17. Performances of different approaches are comparable to those under the Gaussian settings.

### A.3 CITE-SEQ DATASET IMPLEMENTATION DETAILS

We consider the bone marrow CITE-seq dataset from Stuart et al. (2019) which consists of measurements in two modalities on 30672 individual cells: transcriptome (RNA) and 25 cell-surface proteins (ADT). We randomly split the dataset into a training set of size 15000 and a test set of size 15672. We follow the preprocessing in `https://satijalab.org/seurat/articles/weighted_nearest_neighbor_analysis`, where we normalize ADT data with centering

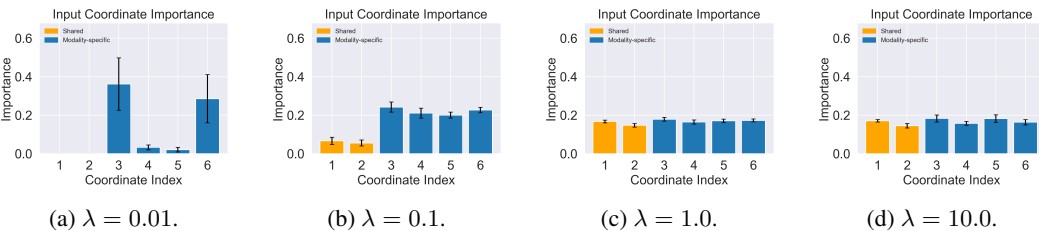

Figure 12: IndiSeek with varying $\lambda$: feature importance averaged over 10 simulations (Setting 3).

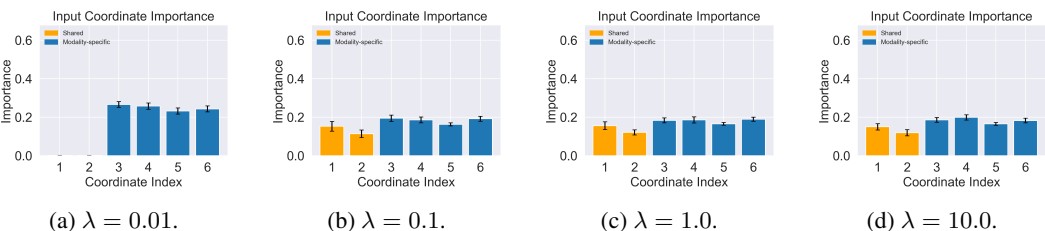

Figure 13: Factorized CL with varying $\lambda$: feature importance averaged over 10 simulations (Setting 3).

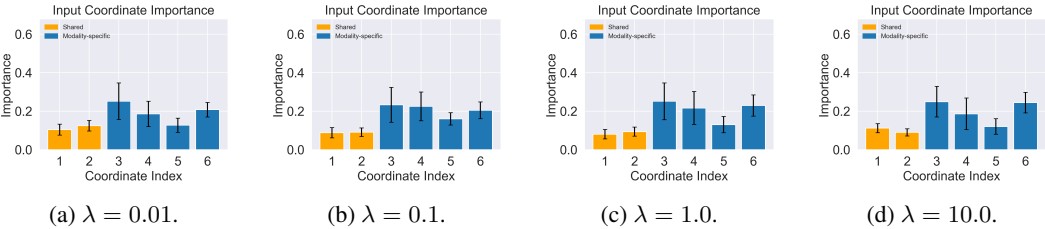

Figure 14: InfoDisen with varying $\lambda$: feature importance averaged over 10 simulations (Setting 3).

to produce a 24-dimensional input based on moments estimated from the training set only, and normalize the high-dimensional RNA data and extract the first 200 principal components as the inputs for CLIP.

In the experiments presented in Figure 4 and Table 5, architectures of representation maps $g$ and $h$ for three methods are all 5-layer ReLU networks with width 50 and output dimension 50. Throughout experiments with real-world datasets, we adopt a batch size of 128 and a learning rate of $10^{-4}$ for training, in which we set the number of epochs to 2000.

In Figure 18, Figure 19, and Figure 20, we visualize the learned representations $(C_i, Z_i)$ by all three methods for each modality. The top rows correspond to the RNA modality, and the bottom rows to the ADT modality. In the left panels, the representations are colored by level-1 cell types, and in the right panels, to ease visualization, we plot representations of a subset of cells whose level-2 cell types are among the top-40% in the average benchmark weights across all level-2 cell types in the corresponding modality. In other words, the chosen level-2 cell types for each modality are those for which the corresponding modality is more important for their distinction. If a disentangled learning approach works well, the resulting modality-specific representations of the chosen level-2 cell types are expected to be well separated in the respective UMAPs. Additionally, in Figure 21, Figure 22, and Figure 23, we visualize the concatenated representations $(C_1, C_2, Z_1, Z_2)$ learned by all three methods, in which representations are colored by level-1 and level-2 cell types in the left and the right panels, respectively. Here, we choose the optimal $\lambda$ implied by Figure 5 for each method, respectively: 0.1 for FactorizedCL, 1.0 for InfoDisen, and 10.0 for IndiSeek.

To quantify the separation of cell types, we adopt two metrics: (1) the Adjusted Rand Index (ARI) (Hubert & Arabie, 1985), which measures agreement between $k$-means clustering on the learned representations and the ground-truth labels, with values ranging from $-1$ to $1$ where higher

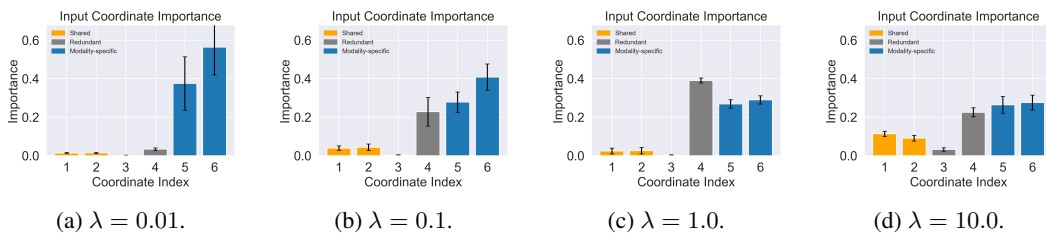

Figure 15: IndiSeek with varying $\lambda$: feature importance averaged over 10 simulations (Setting 4).

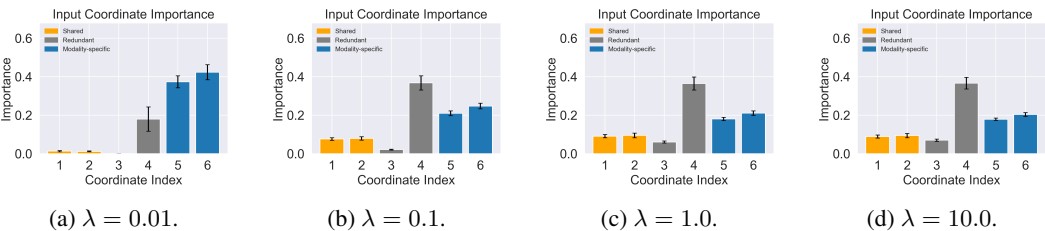

Figure 16: Factorized CL with varying $\lambda$: feature importance averaged over 10 simulations (Setting 4).

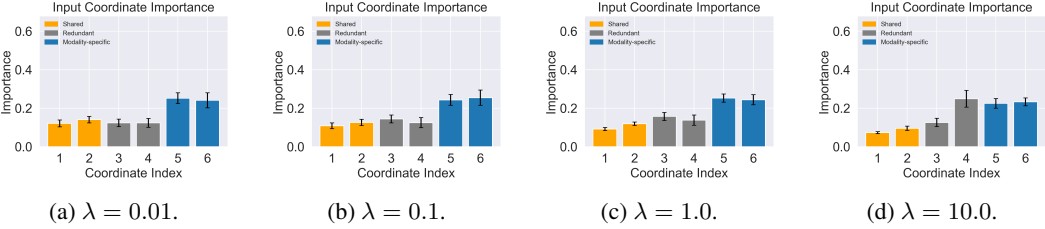

Figure 17: InfoDisen with varying $\lambda$: feature importance averaged over 10 simulations (Setting 4).

values indicate better cluster recovery; and (2) NMI (normalized mutual information) following (Luecken et al., 2022; Pedregosa et al., 2011), which quantifies the mutual dependence between $k$-means clustering and the ground-truth labels, with values in $[0, 1]$ where higher scores indicate better alignment between the discovered clusters and true labels. Figures 18-23 demonstrate that, when measured by these two metrics, IndiSeek outperforms the two SOTA methods in both learned modality-specific representations and the overall learned representations.

### A.3.1 CITE-SEQ RESULTS WITHOUT DIMENSION REDUCTION OF RNA DATA

We also present the results with CITE-seq data following the preprocessing in https://satijalab.org/seurat/articles/weighted_nearest_neighbor_analysis, where we normalize ADT data with centering to produce a 24-dimensional input based on moments estimated from the training set only, and normalize the extremely high-dimensional RNA data. Here, instead of extracting the first 200 principal components as the inputs for CLIP, we adopt the raw normalized 2000-dimensional RNA data for training. The results are presented in Figure 24 and Figure 25, respectively. Compared with Figures 4-5, results with and without the PCA dimension reduction for the RNA modality are comparable.

### A.4 MULTIBENCH DATASETS

We evaluate the performance of our method on MultiBench datasets Liang et al. (2021), including video datasets (MOSI (Zadeh et al., 2016), MOSEI (Zadeh et al., 2018), UR-FUNNY (Hasan et al., 2019), MUStARD (Castro et al., 2019)) with modalities of text, image, and audio, and the

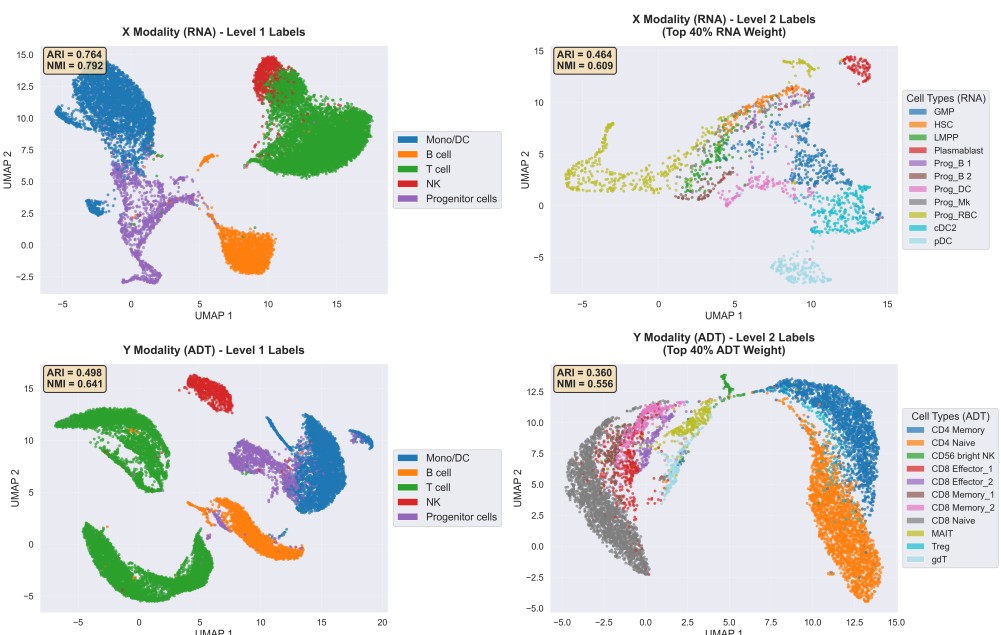

Figure 18: UMAP of IndiSeek-learned representations $(C_i, Z_i)$ (colored by cell types).

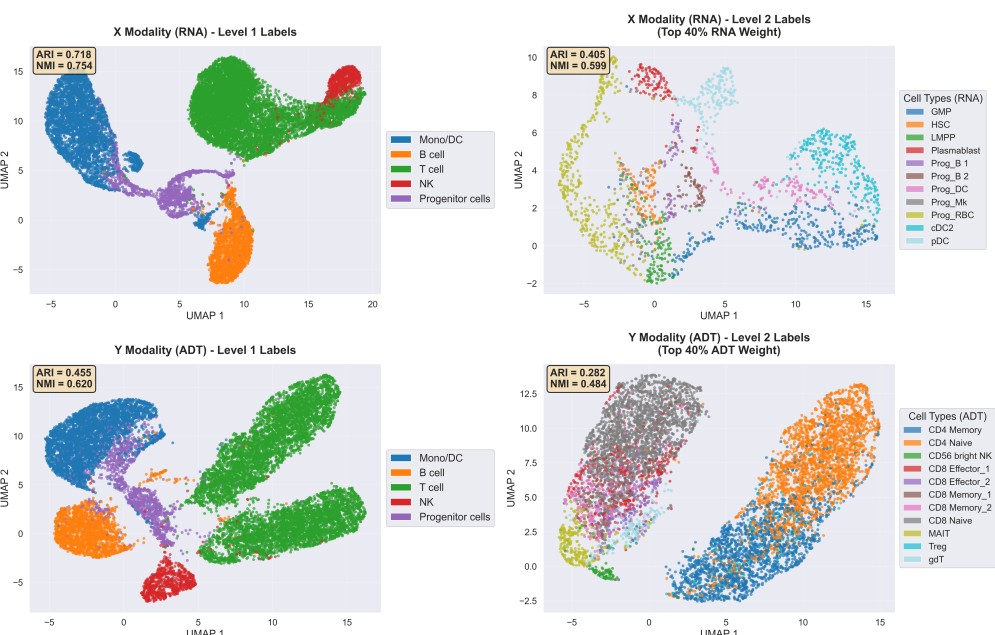

Figure 19: UMAP of FactorizedCL-learned representations $(C_i, Z_i)$ (colored by cell types).

MIMIC dataset (Johnson et al., 2016). For video datasets, following Liang et al. (2023a); Wang et al. (2024a), we focus solely on vision and text modalities, excluding audio data in these examples. More concretely,

- **MOSI** Zadeh et al. (2016): 2199 YouTube clips with vision and text modalities. Task: binary sentiment classification.

- **MOSEI** (Zadeh et al., 2018): 23000 monologue videos with vision and text modalities. Task: binary sentiment classification.

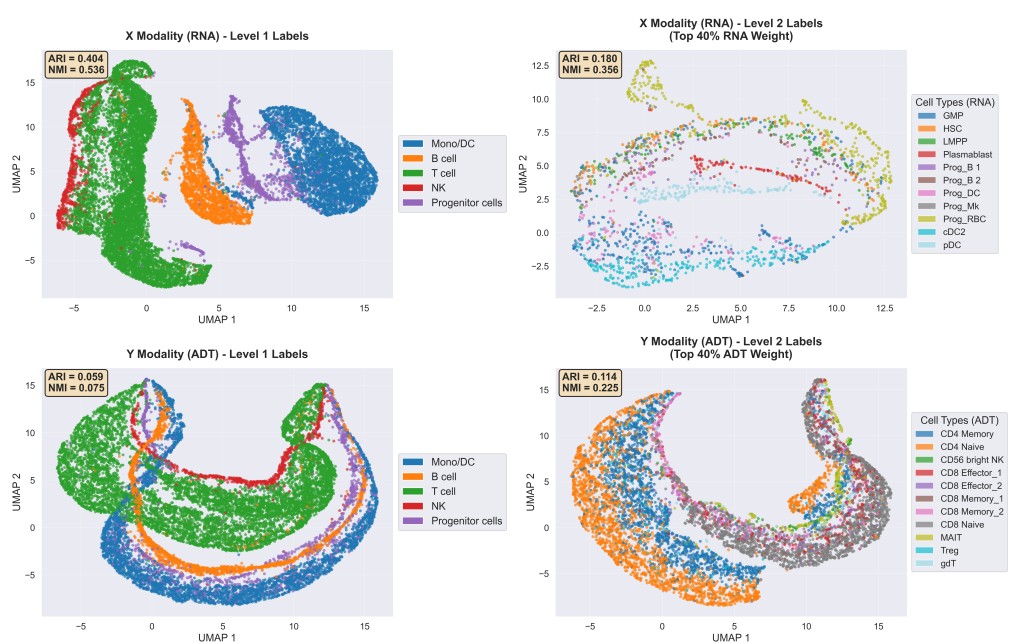

Figure 20: UMAP of InfoDisen-learned representations $(C_i, Z_i)$ (colored by cell types).

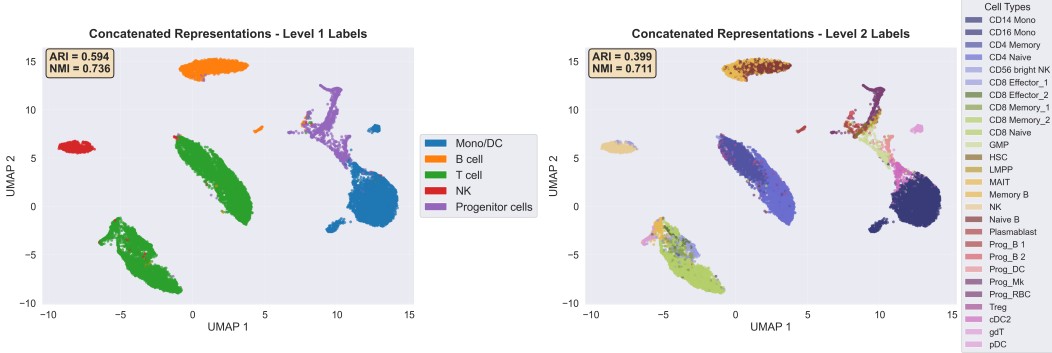

Figure 21: UMAP of concatenated IndiSeek-learned representations $(C_1, C_2, Z_1, Z_2)$ (colored by cell types).

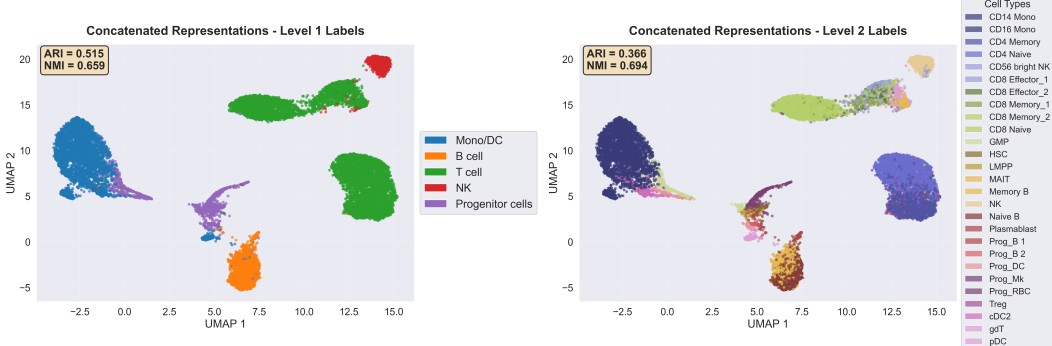

Figure 22: UMAP of concatenated FactorizedCL-learned representations $(C_1, C_2, Z_1, Z_2)$ (colored by cell types).

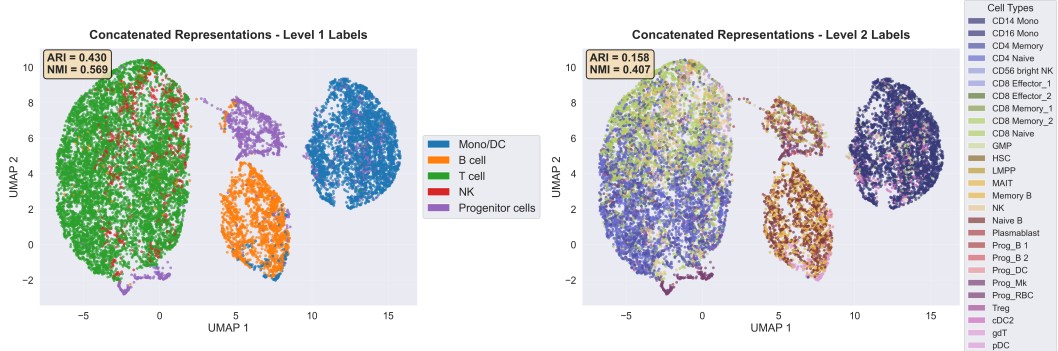

Figure 23: UMAP of concatenated InfoDisen-learned representations $(C_1, C_2, Z_1, Z_2)$ (colored by cell types).

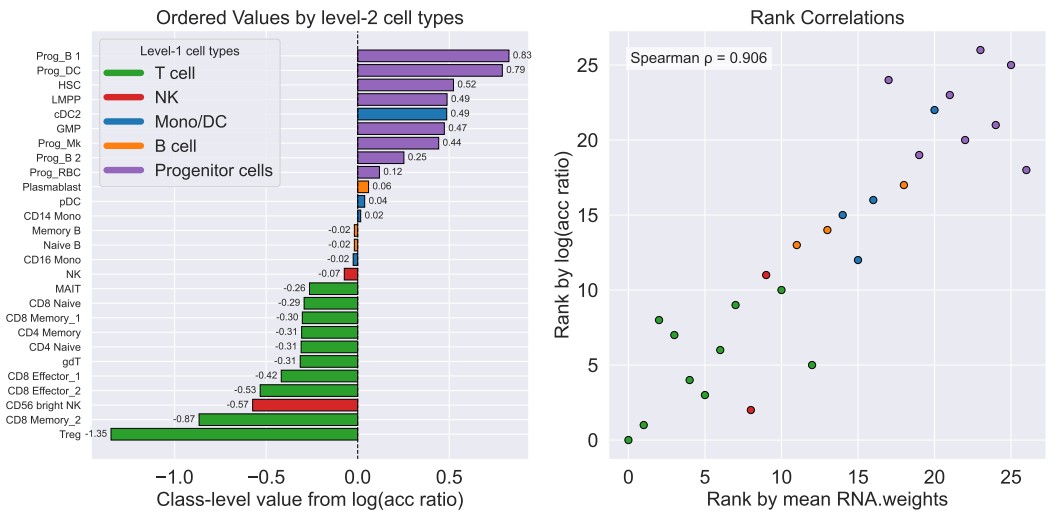

Figure 24: Performance of IndiSeek in CITE-seq dataset ($\lambda = 10.0$).

- **UR-FUNNY** (Hasan et al., 2019): TED talk videos with vision and text modalities. Task: binary humor detection.

- **MUStARD** (Castro et al., 2019): 690 TV show clips with vision and text modalities. Task: binary sarcasm detection.

- **MIMIC** (Johnson et al., 2016): over 36000 ICU patient records with time-series vitals and tabular data. Task: Following Wang et al. (2024a), we aim at predicting ICD-9 group 7 membership.

We follow the processing pipeline in Liang et al. (2023a) and adopt exactly the same data splitting and feature pre-extraction. We use a smaller Transformer architecture with 2 heads, 2 layers, and the intermediate layers with a width 128.

### A.4.1 EXPERIMENTAL RESULTS WITH VARYING $\lambda$

In Section 5, we present the accuracies averaged over 10 seeds with the optimal $\lambda$ for each method. In the following tables, we present more detailed results on the accuracy for each method with different choices of $\lambda$: Table 5 for MOSEI, Table 4 for MOSI, Table 7 for MUStARD, Table 6 for UR-FUNNY, and Table 8 for MIMIC.

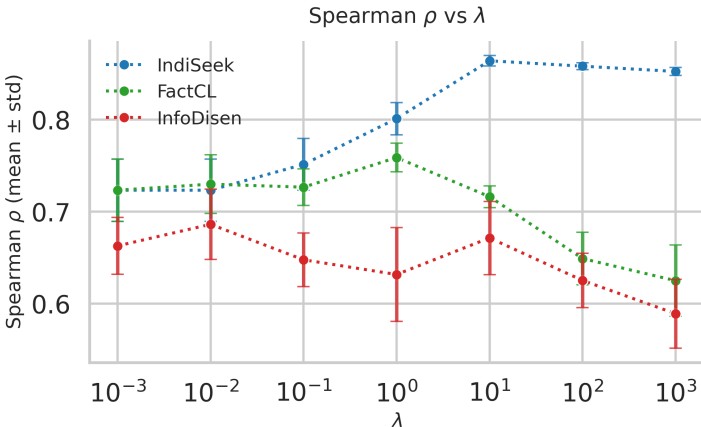

Figure 25: Comparison of rank correlation metrics across three methods on the CITE-seq dataset.

Table 4: MOSI results with varying $\lambda$ (averaged over 10 seeds, standard errors in parentheses). All values are percentages.

| $\lambda$ | 0.01 | 0.1 | 1.0 | 10.0 | 100.0 | 1000.0 | max over $\lambda$ |
|---|---|---|---|---|---|---|---|
| IndiSeek | $67.46_{(0.74)}$ | $69.48_{(0.86)}$ | $70.03_{(1.39)}$ | $66.43_{(0.34)}$ | $65.95_{(0.45)}$ | $65.90_{(0.89)}$ | $70.03_{(1.39)}$ |
| FactorizedCL | $66.76_{(0.58)}$ | $65.07_{(0.92)}$ | $66.12_{(0.67)}$ | $64.26_{(1.07)}$ | $65.07_{(0.58)}$ | $65.12_{(0.89)}$ | $67.11_{(0.34)}$ |
| InfoDisen | $65.57_{(0.84)}$ | $65.31_{(1.26)}$ | $65.83_{(1.00)}$ | $65.22_{(1.08)}$ | $64.84_{(1.26)}$ | $67.52_{(0.62)}$ | $67.52_{(0.62)}$ |
| CLIP | $67.61_{(0.66)}$ | $67.61_{(0.66)}$ | $67.61_{(0.66)}$ | $67.61_{(0.66)}$ | $67.61_{(0.66)}$ | $67.61_{(0.66)}$ | $67.61_{(0.66)}$ |

### A.4.2 TRAINING TIME COMPARISON

We compare the computational efficiency of different methods by measuring the training time required to compute modality-specific features. Specifically, we record the time for training the disentangled representations for both modalities (text and vision) after the initial CLIP feature extraction. All methods are trained with consistent configurations: we use a batch size of 128, a maximum of 2000 epochs with early stopping, Adam optimizer with learning rate $10^{-4}$ and weight decay $10^{-4}$, and Transformer-based encoders to map raw sequences to a 50-dimensional latent space. The training is performed on a single GPU, and we measure the wall-clock time for the complete training of both modality-specific encoders.

Table 9 presents the training time comparison across datasets, averaged over 10 random seeds. The results show that all three methods have comparable computational costs, with differences primarily stemming from dataset sizes and sequence lengths. IndiSeek demonstrates competitive efficiency while achieving superior accuracy (as shown in Table 1), making it a practical choice for multimodal learning tasks.

### A.4.3 ABLATION STUDIES AND COMPARISON WITH COMM

To understand the contribution of individual components in IndiSeek, we conduct an ablation study on the MOSI dataset. Specifically, we compare IndiSeekmethod against IndiSeek0, a variant that removes the modality-switching mechanism and uses a fixed pairing between raw inputs and CLIP embeddings. We also include a comparison with CoMM (Dufumier et al., 2024). Table 10 presents the results of this analysis. CoMM achieves competitive performance at 69.56%, showing that collaborative learning approaches can be effective for multimodal representation learning.

### A.4.4 MULTI-TASK PERFORMANCE ON MIMIC DATASET

In this section, we evaluate IndiSeek on all 20 tasks of predicting ICD-9 groups, and results are summarized in the following table.

Table 5: MOSEI results with varying $\lambda$ (averaged over 10 seeds, standard errors in parentheses). All values are percentages.

| $\lambda$ | 0.01 | 0.1 | 1.0 | 10.0 | 100.0 | 1000.0 | max over $\lambda$ |
|---|---|---|---|---|---|---|---|
| IndiSeek | $74.70_{(0.06)}$ | $74.71_{(0.05)}$ | $75.47_{(0.13)}$ | $75.29_{(0.09)}$ | $75.19_{(0.11)}$ | $75.03_{(0.11)}$ | $75.47_{(0.13)}$ |
| FactorizedCL | $74.74_{(0.04)}$ | $74.74_{(0.05)}$ | $74.65_{(0.04)}$ | $74.69_{(0.07)}$ | $74.65_{(0.04)}$ | $74.60_{(0.06)}$ | $74.74_{(0.04)}$ |
| InfoDisen | $74.65_{(0.06)}$ | $74.61_{(0.03)}$ | $74.58_{(0.04)}$ | $74.70_{(0.03)}$ | $74.73_{(0.08)}$ | $74.56_{(0.04)}$ | $74.73_{(0.08)}$ |
| CLIP | $74.70_{(0.05)}$ | $74.70_{(0.05)}$ | $74.70_{(0.05)}$ | $74.70_{(0.05)}$ | $74.70_{(0.05)}$ | $74.70_{(0.05)}$ | $74.70_{(0.05)}$ |

Table 6: UR-FUNNY results with varying $\lambda$ (averaged over 10 seeds, standard errors in parentheses). All values are percentages.

| $\lambda$ | 0.01 | 0.1 | 1.0 | 10.0 | 100.0 | 1000.0 | max over $\lambda$ |
|---|---|---|---|---|---|---|---|
| IndiSeek | $58.85_{(1.08)}$ | $62.40_{(0.82)}$ | $63.08_{(0.44)}$ | $63.79_{(0.39)}$ | $63.12_{(0.55)}$ | $63.71_{(0.62)}$ | $63.79_{(0.39)}$ |
| FactorizedCL | $58.36_{(0.25)}$ | $57.81_{(0.41)}$ | $58.22_{(0.38)}$ | $57.43_{(0.64)}$ | $57.13_{(0.24)}$ | $57.02_{(0.32)}$ | $58.36_{(0.25)}$ |
| InfoDisen | $56.09_{(0.33)}$ | $56.14_{(0.98)}$ | $56.28_{(0.76)}$ | $57.52_{(0.57)}$ | $58.08_{(0.50)}$ | $57.73_{(0.70)}$ | $58.08_{(0.50)}$ |
| CLIP | $58.32_{(0.68)}$ | $58.32_{(0.68)}$ | $58.32_{(0.68)}$ | $58.32_{(0.68)}$ | $58.32_{(0.68)}$ | $58.32_{(0.68)}$ | $58.32_{(0.68)}$ |

In Figure 11, IndiSeek leads the performance in most tasks. Moreover, in many real-world scenarios, downstream tasks are unknown in advance, or learned representations are expected to handle hundreds of tasks simultaneously. In this case, the task-relevant optimality of augmentations could be barely met in Liang et al. (2023a), which highlights the flexibility and practicality of task-agnostic disentangled learning.

Table 7: MUStARD results with varying $\lambda$ (averaged over 10 seeds, standard errors in parentheses). All values are percentages.

| $\lambda$ | 0.01 | 0.1 | 1.0 | 10.0 | 100.0 | 1000.0 | max over $\lambda$ |
|---|---|---|---|---|---|---|---|
| IndiSeek | $55.51_{(1.28)}$ | $57.46_{(1.04)}$ | $55.72_{(1.95)}$ | $55.07_{(1.06)}$ | $53.48_{(1.42)}$ | $52.25_{(1.17)}$ | $57.46_{(1.04)}$ |
| FactorizedCL | $53.91_{(0.57)}$ | $54.71_{(1.21)}$ | $54.86_{(1.10)}$ | $54.64_{(0.70)}$ | $56.45_{(1.16)}$ | $55.07_{(0.91)}$ | $56.45_{(1.16)}$ |
| InfoDisen | $54.13_{(0.83)}$ | $53.41_{(1.00)}$ | $54.42_{(0.97)}$ | $54.57_{(0.94)}$ | $56.16_{(0.92)}$ | $55.58_{(1.27)}$ | $56.16_{(0.92)}$ |
| CLIP | $55.36_{(1.12)}$ | $55.36_{(1.12)}$ | $55.36_{(1.12)}$ | $55.36_{(1.12)}$ | $55.36_{(1.12)}$ | $55.36_{(1.12)}$ | $55.36_{(1.12)}$ |

Table 8: MIMIC results with varying $\lambda$ (averaged over 10 seeds, standard errors in parentheses). All values are percentages.

| $\lambda$ | 0.01 | 0.1 | 1.0 | 10.0 | 100.0 | 1000.0 | max over $\lambda$ |
|---|---|---|---|---|---|---|---|
| IndiSeek | $65.52_{(0.22)}$ | $65.79_{(0.17)}$ | $65.82_{(0.24)}$ | $65.89_{(0.24)}$ | $65.96_{(0.28)}$ | $65.99_{(0.31)}$ | $65.99_{(0.31)}$ |
| FactorizedCL | $65.34_{(0.29)}$ | $65.11_{(0.31)}$ | $65.69_{(0.11)}$ | $65.33_{(0.30)}$ | $64.92_{(0.35)}$ | $65.24_{(0.39)}$ | $65.69_{(0.11)}$ |
| InfoDisen | $64.89_{(0.30)}$ | $64.80_{(0.31)}$ | $65.10_{(0.25)}$ | $64.96_{(0.38)}$ | $65.47_{(0.23)}$ | $65.32_{(0.24)}$ | $65.47_{(0.23)}$ |
| CLIP | $64.56_{(0.29)}$ | $64.56_{(0.29)}$ | $64.56_{(0.29)}$ | $64.56_{(0.29)}$ | $64.56_{(0.29)}$ | $64.56_{(0.29)}$ | $64.56_{(0.29)}$ |

Table 9: Comparison of training time on multimodal datasets (averaged over 10 seeds, standard errors in parentheses, max average over $\lambda$). All values are in seconds ($\times 10^2$).

| Method | MOSI | MOSEI | UR-FUNNY | MUStARD | MIMIC |
|---|---|---|---|---|---|
| IndiSeek | $7.11_{(0.09)}$ | $81.43_{(0.83)}$ | $40.75_{(0.29)}$ | $2.63_{(0.04)}$ | $132.74_{(1.76)}$ |
| FactorizedCL | $7.35_{(0.09)}$ | $83.27_{(0.87)}$ | $41.97_{(0.31)}$ | $2.72_{(0.04)}$ | $135.93_{(1.84)}$ |
| InfoDisen | $7.18_{(0.09)}$ | $81.32_{(0.85)}$ | $40.70_{(0.30)}$ | $2.65_{(0.04)}$ | $132.60_{(1.98)}$ |

Table 10: MOSI ablation study results with varying $\lambda$ (averaged over 10 seeds, standard errors in parentheses). All values are percentages.

| $\lambda$ | 0.01 | 0.1 | 1.0 | 10.0 | 100.0 | 1000.0 | max over $\lambda$ |
|---|---|---|---|---|---|---|---|
| IndiSeek | $67.11_{(0.85)}$ | $69.61_{(0.55)}$ | $70.09_{(0.80)}$ | $66.63_{(0.56)}$ | $66.43_{(0.50)}$ | $66.06_{(0.72)}$ | $70.09_{(0.80)}$ |
| IndiSeek0 | $66.24_{(0.43)}$ | $69.21_{(0.48)}$ | $69.59_{(0.27)}$ | $66.57_{(0.49)}$ | $65.73_{(0.35)}$ | $65.07_{(0.21)}$ | $69.59_{(0.27)}$ |
| CoMM | $68.34_{(0.79)}$ | $68.69_{(0.65)}$ | $69.56_{(0.47)}$ | $67.14_{(0.37)}$ | $66.79_{(0.59)}$ | $66.60_{(0.44)}$ | $69.56_{(0.47)}$ |
| FactorizedCL | $66.76_{(0.41)}$ | $65.07_{(0.65)}$ | $66.12_{(0.47)}$ | $64.26_{(0.76)}$ | $65.07_{(0.41)}$ | $65.12_{(0.63)}$ | $67.11_{(0.24)}$ |
| InfoDisen | $65.57_{(0.59)}$ | $65.31_{(0.89)}$ | $65.83_{(0.71)}$ | $65.22_{(0.76)}$ | $64.84_{(0.89)}$ | $67.52_{(0.44)}$ | $67.52_{(0.44)}$ |
| CLIP | $67.14_{(0.87)}$ | $67.14_{(0.87)}$ | $67.14_{(0.87)}$ | $67.14_{(0.87)}$ | $67.14_{(0.87)}$ | $67.14_{(0.87)}$ | $67.14_{(0.87)}$ |

Table 11: Multi-task performance on MIMIC dataset (averaged over 10 seeds, standard errors in parentheses, max average over $\lambda$). All values are percentages.

| Task ID | IndiSeek | FactorizedCL | InfoDisen | CLIP |
|---------|----------|--------------|-----------|------|
| 0 | $77.73_{(0.06)}$ | $77.76_{(0.10)}$ | $77.34_{(0.04)}$ | $77.07_{(0.18)}$ |
| 1 | $91.31_{(0.03)}$ | $91.30_{(0.02)}$ | $91.30_{(0.02)}$ | $91.12_{(0.07)}$ |
| 2 | $70.76_{(0.19)}$ | $70.67_{(0.11)}$ | $70.62_{(0.24)}$ | $70.22_{(0.09)}$ |
| 3 | $67.04_{(0.10)}$ | $67.01_{(0.08)}$ | $66.78_{(0.12)}$ | $66.89_{(0.11)}$ |
| 4 | $71.04_{(0.22)}$ | $70.86_{(0.32)}$ | $70.87_{(0.20)}$ | $70.81_{(0.07)}$ |
| 5 | $72.22_{(0.08)}$ | $72.03_{(0.05)}$ | $72.11_{(0.09)}$ | $71.97_{(0.09)}$ |
| 6 | $86.00_{(0.06)}$ | $86.17_{(0.10)}$ | $85.94_{(0.11)}$ | $85.98_{(0.07)}$ |
| 7 | $66.04_{(0.18)}$ | $65.81_{(0.35)}$ | $65.58_{(0.34)}$ | $64.50_{(0.42)}$ |
| 8 | $65.66_{(0.07)}$ | $65.84_{(0.05)}$ | $65.50_{(0.19)}$ | $65.04_{(0.12)}$ |
| 9 | $73.09_{(0.17)}$ | $73.16_{(0.09)}$ | $72.75_{(0.19)}$ | $72.44_{(0.19)}$ |
| 10 | $99.60_{(0.00)}$ | $99.60_{(0.01)}$ | $99.60_{(0.00)}$ | $99.60_{(0.00)}$ |
| 11 | $89.14_{(0.00)}$ | $89.14_{(0.00)}$ | $89.14_{(0.00)}$ | $89.14_{(0.00)}$ |
| 12 | $80.93_{(0.00)}$ | $80.94_{(0.02)}$ | $80.93_{(0.01)}$ | $80.93_{(0.00)}$ |
| 13 | $96.69_{(0.00)}$ | $96.69_{(0.02)}$ | $96.69_{(0.01)}$ | $96.69_{(0.00)}$ |
| 14 | $69.43_{(0.05)}$ | $69.21_{(0.17)}$ | $69.13_{(0.13)}$ | $68.99_{(0.02)}$ |
| 15 | $90.98_{(0.00)}$ | $90.98_{(0.00)}$ | $90.98_{(0.00)}$ | $90.98_{(0.00)}$ |
| 16 | $96.84_{(0.00)}$ | $96.84_{(0.00)}$ | $96.84_{(0.00)}$ | $96.84_{(0.00)}$ |
| 17 | $62.58_{(0.17)}$ | $61.83_{(0.20)}$ | $61.72_{(0.14)}$ | $61.50_{(0.14)}$ |
| 18 | $60.99_{(0.19)}$ | $60.36_{(0.14)}$ | $60.40_{(0.34)}$ | $60.71_{(0.09)}$ |
| 19 | $69.71_{(0.11)}$ | $69.30_{(0.12)}$ | $69.46_{(0.14)}$ | $69.25_{(0.16)}$ |
| Average | 77.89 | 77.82 | 77.74 | 77.55 |

