# OpenReview forum: "IndiSeek learns information-guided disentangled representations"
_ICLR.cc/2026/Conference — ICLR 2026 Conference Withdrawn Submission_

### Official Review · Reviewer_gHk3 · 2025-10-21

**Soundness:** 3
**Presentation:** 2
**Contribution:** 3
**Rating:** 4
**Confidence:** 3

**Summary:**

This paper presents IndiSeek, a novel framework for disentangled representation learning in multi-modal settings. The goal is to extract shared features that capture cross-modal information and modality-specific features that retain complementary information unique to each modality. The method builds on an information-theoretic formulation that balances independence (minimizing mutual information between shared and specific components) and completeness (maximizing the retained information about the input).

**Strengths:**

- Derives a tractable objective from mutual information principles, avoiding fragile variational estimations.

- Comprehensive experiments (synthetic, biological, and benchmark) with consistent outperformance of SOTA baselines.

- Visualizations (Figures 1–2, 4) and correlation analyses (Figure 5) confirm that IndiSeek captures meaningful modality-specific patterns.

- Works in diverse domains (CITE-seq biology, video, text, audio), demonstrating transferability.

**Weaknesses:**

- The method relies on manual grid search for the tradeoff hyperparameter λ. A principled or adaptive selection scheme would enhance practicality.

- While the reconstruction bound is justified via Fano’s inequality, tightness conditions (e.g., non-Gaussian distributions) are not deeply discussed.

- Experiments could include more recent disentanglement baselines or adversarial approaches (e.g., variational or adversarial mutual information bounds).

- Although efficient in small models, scalability to very high-dimensional modalities (e.g., video) may present computational challenges.

- Results mainly rely on linear classifiers; evaluating transfer learning or generative reconstruction would add depth.

**Questions:**

- How does IndiSeek perform when the reconstruction loss is replaced by more expressive models (e.g., diffusion-based decoders)?

- Can λ be adaptively tuned during training, perhaps via normalization of CLUB and reconstruction magnitudes?

- What is the computational cost of CLUB estimation relative to InfoNCE in larger settings?

- Could the framework handle partial modality observation, e.g., missing one modality at inference?

- How sensitive are results to the dimensionality of shared vs. specific latent spaces?

**Details Of Ethics Concerns:**

The datasets are publicly available, and no sensitive human data or ethical concerns are present. The paper’s application to biological data respects established standards for reproducibility and transparency.

---

> ### Author Response · Authors · 2025-11-24
>
> [**Choice of $\lambda$**] Thanks for the question on the selection of this hyperparameter, which is a crucial ingredient in our approach.
> To better understand the tradeoff between two terms and relieve the artifact in the “optimal” $\lambda$ in various examples, we consider a scale-free version of the reconstruction loss, in which, instead of $\Vert h(C,Z) - X\Vert^2$, we use $s^{-2}(X)\|\|h(C,Z) - X\|\|^2$ where $s^2(X)=\|\|X\|\|^2$. Then, with this scaled reconstruction loss, we rerun the simulations as well as experiments and present the updated results in the draft. With the rescaled reconstruction loss, the CLUB upper bound and the reconstruction term are now both scale-free, and our experimental results have suggested a candidate range of $\lambda$'s between $10^{-1}$ to $10^1$ (as shown in Table 4-8).
>
>
> [**Tightness of MSE**] Thank you for raising the question of the tightness of MSE. It is indeed true that MSE, as an upper bound, can be conservative than mutual information. However, if the data-generating process indeed admits shared and specific features that can be recovered by an expressive function class, the minimum of the reconstruction term $-I(X; C,Z)$ would be $-\infty$ (i.e., the minimum of conditional entropy will be zero), in which sense, minimizing MSE will ensure the reconstruction. Also, as we can see in synthetic examples, where the distribution of features is no longer Gaussian after MLP, IndiSeek with MSE can recover all specific features, which, to some extent, justifies the robustness of MSE as an upper bound for the conditional entropy.
>
> [**CoMM**] We have added additional experimental results which incorporate the comparison with CoMM in [1] in Appendix A.4 Table 10. With the MOSI dataset, we can see that the collaborative mechanism of CoMM can be effective in the downstream task, but IndiSeek still outperforms methods in comparison.
>
> [**Computational cost**] Thank you for pointing out this question on computational cost! In current examples, to demonstrate that IndiSeek may not suffer from computational cost, we have presented the comparison of time spent in training (with the same number of epochs) across all methods. The table has been added to Appendix B.4.2 (Table 9), in which, all three methods have comparable computational costs, with differences primarily stemming from dataset sizes and sequence lengths. IndiSeek demonstrates competitive efficiency while achieving superior accuracy, making it a practical choice for multimodal learning tasks. There could be a tradeoff and balance between the high dimensionality in reconstruction and the simplicity of MSE objective.
>
>
> [**Sensitivity to dimensionality**] Thank you for pointing this out! For the shared features, as is pointed out by some previous works on CLIP, the quality of learned representations is robust to the output dimension when beyond a threshold. We have also added ablation studies in Table 2 with varying output dimensions, from which we can see that the performance of IndiSeek is robust to the output dimension, and across all output dimensions, IndiSeek consistently outperforms in downstream accuracy.
>
> ---
>
> [1] Dufumier, Benoit, Javiera Castillo-Navarro, Devis Tuia, and Jean-Philippe Thiran. "What to align in multimodal contrastive learning?." arXiv preprint arXiv:2409.07402 (2024).
>
> ---
>
> For additional revisions made, please see "Reply to all reviewers" above.

---

### Official Review · Reviewer_4mW7 · 2025-10-29

**Soundness:** 3
**Presentation:** 2
**Contribution:** 2
**Rating:** 4
**Confidence:** 4

**Summary:**

This paper introduces a new combination of objectives to learn disentangled representations for multi-modal data: “IndiSeek” first learns shared features using CLIP, followed by the specific features using NCE-CLUB and a simple reconstruction loss. These two objectives in the second stage ensure disentanglement from the shared information and completeness of the full representation. They show that approximating conditional mutual information with a reconstruction loss is sufficient and performs better than InfoNCE loss used in comparable methods. The authors compare IndiSeek’s performance with two highly comparable SOTA methods on simulations, and various real world datasets.

**Strengths:**

1) The core idea of using CLIP in a 2-stage solution and replacing the InfoNCE loss with reconstruction loss for completeness is simple but intuitive. While the novelty is incremental, the approach is well-justified with information-theoretic bounds and promises to be more robust.

2) The method is tested on a wide variety of data (simulation, biological data, multibench datasets), which demonstrates its applicability to many domains.

**Weaknesses:**

1) There seems to be a problem of information leakage from the modality-specific information into the shared representations C during the first phase of training with CLIP. Information from more than just shared variables can easily leak into C via CLIP. And since it seems like C is not updated during the second phase, there is a high chance of C containing modality-specific information, which then may not even get into Z because it is not required per the completeness. The paper provides no discussion, proof of disentanglement, or simulation analysis on this issue. Given the focus of the work being disentanglement, more extensive experiments should be conducted related to this.

2) While the paper provides results on many datasets (especially in the case of MultiBench), their significance is unclear. The average accuracies over 10 random seeds are provided without error margins. It is crucial to report for example the SD or SEM to determine if the performance is at all statistically significant. Given that the methodological novelty is incremental, it is important to show that it works better than competitors and is thus relevant.

3) The motivating example (Section 1.1) is presented at the very beginning. As a reader, this was very confusing because the method and the baselines were not yet introduced, making the results hard to appreciate. It would be a lot more effective to move this right after section 3 and merge paper structure into the introduction.

4) The paper does not state the input feature dimension for RNA. Also, the descriptions of used architectures (i.e. “5-layer ReLU NNs with middle layers of width 50”) are also not quite sufficient. What is the hidden dimension? Does this refer to the encoder and the decoder? I am further concerned with the small fixed MLP (width 50) for both the low-dimensional ADT data and the high-dimensional RNA. For one this leads to an over-, for the other to an under-determined model and may skew the results of their relative importance. It would be good to at least have some results on the reconstruction performance on each of the modalities here as a sanity check.

5) The paper only compares against two very similar methods (Factorized CL and InfoDisen). While these are the most relevant, the authors could include other families of multi-modal disentanglement methods, such as VAE-based approaches (e.g. DMVAE, IDMVAE) or alternative information-theoretic methods (e.g. DisentangledSSL). While they discuss why some other methods were not included in the comparison (mainly if they are designed with task-dependence), other works like CoMM still compare to Factorized CL for example and since some MultiBench accuracies are reported on multi-task performance, these could be included too. But I think if the two points above are addressed adequately, this is only a minor weakness since I get the motivation of showing that the 2-stage approach and objective change make a big difference.

**Questions:**

1) Minor formatting and text issues: For example, the error bars in Figure 5 are not explained in text or caption. I think I know what equation 2 is trying to say, but I think there are some latex errors / spelling mistakes?

2) Missing methodological details: the decoder network (g) is not explicitly described in Section 3 and needs to be introduced for clarity and reproducibility.

---

> ### Author Response · Authors · 2025-11-24
>
> - [Imperfectness of shared features] Thanks for this very insightful question! This question is also closely related to questions mentioned by Reviewer hMRU and D7zU.
>     - In (Wang et al., 2024), as the reviewer has mentioned, the objective for modality 1 takes the form $\min - I(Z\_1, C\_2; X\_1) + \lambda \cdot I(Z\_1; C\_1)$ which encourages the independence between $Z\_1$ and $C\_1$ while encouraging $(Z\_1,C\_2)$ to reconstruct $X\_1$. However, this could cause issues as follows. Consider a scenario where there is an underlying shared component $C^\*$ between $X\_1$ and $X\_2$, in which case, with information leakage, we can assume $C\_1 = K\_1(C^\*, X\_1)$ and $C\_2 = K\_2(C^\*, X\_2)$. In this case, if we directly enforce the independence between $C\_1$ and $Z\_1$, it will lead to information loss in learned $Z\_1$, and in the meantime, this $Z\_1$ together with $C_2$ cannot recover $X_1$, which indicates that if we replace $C\_1$ with $C\_2$ in the reconstruction loss, two terms cannot be minimized simultaneously when there is information leakage in $C\_1$.
>     - This very insightful question indeed motivates the modification of our procedure, in which we can consider the objective $\min - I(Z\_1, C\_1; X\_1) + \lambda \cdot I(Z\_1; C\_2)$, i.e., we replace $C\_1$ by $C\_2$ in the disentanglement term to address the information leakage scenario. We have also updated our draft as well as our experiments, and the intuition is as follows. Note that as long as $C\_1 = K\_1(C^\*, X\_1)$ and $C\_2 = K\_2(C^\*, X\_2)$, i.e., the shared feature $C^\*$ is included in each shared representation, once $Z\_1$ is independent of $C^\*$, $I(Z\_1; C\_2) = I(Z\_1;C^*) = 0$ is true. In this case, we will not lose modality-specific information in learning $Z\_1$, which can improve (Wang et al, 2024)’s method in the information leakage setting and can improve the quality of learned specific features when the shared features are imperfect.
>     - We have already updated experiments using the cross-modal shared features, and notably, with the MOSI dataset, which has a very small sample size–in which case, the CLIP learned features can be undesirable due to a limited sample size, the updated algorithm leads to significant improvement in accuracy.
>
>
> - [Error bars] Thank you for pointing this out! We have now added error bars and standard errors for all experimental results. To clarify, in Figure 5, the error bars are for results over 10 random seeds as well. Additionally, motivated by the previous question and questions of Reviewer hMRU, we have improved IndiSeek via implementing (1) cross-modal disentanglement objective to address the imperfection of shared features; and (2) a rescaled reconstruction loss to balance the magnitude of two terms. We have updated our draft to implement these changes, in which the improvement of IndiSeek is more pronounced across datasets.
>
> - [Organization of examples] Thank you for your suggestion on the paper outline. We will incorporate your advice in polishing our updated draft!
>
> - [Architectures and data dimension] We thank the reviewer for pointing this out.
> For the CITE-seq bone marrow dataset, we follow the preprocessing in \url{https://satijalab.org/seurat/articles/weighted_nearest_neighbor_analysis}, where we normalize ADT data with centering to produce a 24-dimensional input based on moments estimated from the training set only, and normalize the extremely high-dimensional RNA data and extract the first 200 principal components as the inputs for CLIP.
> For the architecture of g, in each experiment, the architecture of $g$ always matches the family of the architecture of h but with different input and output dimensions. Concretely, the input of g is $d\_Z + d\_C$, and the output is $d\_X$; with either MLP or Transformer architectures, the middle layer width of g remains the same as that of $h$.
>
> - [CoMM] We have added additional experimental results which incorporate the comparison with CoMM in [1] in Appendix A.4 Table 10. With the MOSI dataset, we can see that the collaborative mechanism of CoMM can be effective in the downstream task, but IndiSeek still outperforms methods in comparison.
>
> ---
> [1] Dufumier, Benoit, Javiera Castillo-Navarro, Devis Tuia, and Jean-Philippe Thiran. "What to align in multimodal contrastive learning?." arXiv preprint arXiv:2409.07402 (2024).

---

> > ### Comment · Reviewer_4mW7 · 2025-11-27
> >
> > I thank the authors for their detailed response, the additional results (error bars, CoMM comparison), and the clarifications regarding the objective function and data preprocessing. While I appreciate these efforts, a few questions and concerns remain:
> >
> > Information leakage: the clarification provided by the authors help, but I still have some questions:
> > Isn’t Equation 1 wrong then if that is supposed to show the optimization from Wang et al. (also, the citation is missing in the new version)?
> > How are you ensuring independence btw. / no leakage from $Z_1$ into $C_1$ if you are minimizing $I(Z_2;C_1)$?
> > Is C still being optimized in step 2 (3.2)? Equation (3) looks like only Z is optimized. But in step 1 using CLIP, there can definitely be redundancy (modality-specific information) in C.
> >
> > CITE-seq data and real data in general: Reducing RNA data to 200 PCs is a very strong dimensionality reduction. Working with only a handful dimensions from ADT and then tens of thousands from RNA is a challenging application, but it would have been interesting to see if the method can be applied to high-dimensional data. Looking closer, I believe the authors should emphasize early on and in the discussion that they evaluated IndiSEEK only on precomputed embeddings (aside from low-dimensional ADT), which in my eyes makes it less of a representation learning and more of a decomposition method.

---

> ### Author Response · Authors · 2025-11-28
>
> We thank the reviewer for timely feedback, and we would like to address your concerns as follows. For additional revisions made, please see "Reply to all reviewers" above.
>
> - [Preprocessing of CITE-seq] Thanks for your insightful question. To address your concern about using top-200 PCs as the RNA input, we also rerun the experiments with the raw normalized RNA data, which is 2000-dimensional, following https://satijalab.org/seurat/articles/weighted_nearest_neighbor_analysis. We have presented the results in Appendix A 3.1, Figures 24 and 25, from which we can see that the results are very similar to those using top-200 PCs as the RNA input, and the advantage of IndiSeek can also be demonstrated in comparison with baselines in Figure 25.
>
> - [Shared features] We would like to clarify that the goal is not to avoid leakage from $C\_1$ to $Z\_1$, but rather the underlying shared information to $Z\_1$. Also, we would like to clarify that IndiSeek is a two-step procedure, and after Step 1, shared features are fixed in the following step. If Step 1 is successful, $C\_2$ contains all the information shared between $X\_1$ and $X\_2$, and any additional information to the shared part is only from $X\_2$, not from $X_1$. Thus, by minimizing $I(Z\_1; C\_2)$, we are effectively minimizing $I(Z\_1; \text{shared})$, which is more efficient since it is actually possible that there is some redundancy between $C\_1$ and $Z\_1$ as both are functions of $X\_1$. Additionally, the contribution of IndiSeek mainly lies in the disentanglement step, which actually leaves high flexibility in the choice of algorithms (e.g., CLIP, which is shown probably to be maximizing mutual information between modalities [1,2]) in learning shared features in Step 1.
>
> - [Shared features - an additional example] Consider a simple example where $X\_1 = (C^* + \sin(\xi), Z^*\_1)$ and $X\_2 = (C^\*, Z^\*\_2)$ where $C^\*$, $Z^\*\_1$, $Z^\*\_2$, and $\xi$ are mutually independent standard Gaussian variables. In this case, shared features $C\_1=f\_1(X\_1)$ and $C\_2 = f\_2(X\_2)$ cannot achieve exact alignment, i.e., when maximizing mutual information between $C\_1$ and $C\_2$, it is inevitable to include additional redundancy brought by $\xi$ since $C^\*$ can only enter $C\_1$ together with $\xi$.  Now, minimizing $I(Z\_1, C\_1)$ for disentanglement learning in modality 1 will lead to loss of information in $\xi$ in the learned modality-specific feature, while replacing $C\_1$ with $C\_2$ resolves this issue (Section 3.2).
>
>
> ---
> [1] Oko, Kazusato, Licong Lin, Yuhang Cai, and Song Mei. "A statistical theory of contrastive pre-training and multimodal generative ai." arXiv preprint arXiv:2501.04641 (2025).
>
> [2] Oord, Aaron van den, Yazhe Li, and Oriol Vinyals. "Representation learning with contrastive predictive coding." arXiv preprint arXiv:1807.03748 (2018).

---

### Official Review · Reviewer_D7zU · 2025-10-29

**Soundness:** 3
**Presentation:** 3
**Contribution:** 3
**Rating:** 4
**Confidence:** 4

**Summary:**

- The paper tackles disentangling shared versus modality-specific information in multimodal data and proposes IndiSeek, which first learns shared features with a standard contrastive stage and then learns modality-specific features by jointly enforcing independence from the shared part and accurate reconstruction of each modality.
- The method replaces intractable mutual-information objectives with two surrogates: an NCE-CLUB term to upper-bound dependence between shared and specific parts and a reconstruction loss that upper-bounds the conditional entropy, thereby encouraging completeness of the modality-specific information.
- The approach is validated on controlled simulations, on a CITE-seq single-cell dataset where learned specific features align with known RNA-versus-protein importance, and on MultiBench where linear probes over the learned representations outperform baselines on several tasks.

**Strengths:**

- The paper frames disentanglement with a clear two-goal target: independence from the shared features and completeness of modality-specific information, and then instantiates each goal with a surrogate that is consistent with the desired optimization direction.
- The simulations are set up to isolate modality-specific signal and show where prior methods either over-entangle or miss relevant coordinates, while IndiSeek recovers the correct factors across multiple regimes and hyperparameters.
- The paper is well written and is easy to follow.

**Weaknesses:**

- The independence term relies on NCE-CLUB, whose tightness and variance can be sensitive to density-ratio estimation quality, so residual dependence between shared and specific parts could persist and re-enter through the reconstruction path.
- The contrastive stage assumes that the learned shared representation is close to minimally sufficient, but if the contrastive training under- or over-captures cross-modal information, the second stage may respectively force specific features to absorb shared signal or collapse useful unique signal.
- The MultiBench gains are mostly modest and while are reported as best over $\lambda$ and averaged over seeds but without confidence intervals or error bars, which makes it difficult to assess the practical margin and stability of improvements.
- The linear-probe setup concatenates shared and specific features, so improvements could partly reflect larger feature capacity rather than better disentanglement, and there is no comparison to capacity-matched baselines (for example, CLIP with a projection to the same total dimensionality).

**Questions:**

see weaknesses above

---

> ### Author Response · Authors · 2025-11-24
>
> [**Imperfect shared features**] Thanks for this very insightful question! This question is also closely related to the question mentioned by Reviewer hMRU.
>
> - This insightful question indeed motivates the improvement of our procedure, and we have modified the optimization objective to  $\min - I(Z\_1, C\_1; X\_1) + \lambda \cdot I(Z\_1; C\_2)$, i.e., we replace $C\_1$ by $C\_2$ in the disentanglement term to address the scenario with imperfect shared features. We have also updated our draft as well as all our experiments based on the new formulation.
> - Consider a simple example where $X\_1 = (C^\* + \sin(\xi), Z^\*\_1)$ and $X\_2 = (C^\*, Z^*\_2)$ where $C^\*$, $Z^\*\_1$, $Z^\*\_2$, and $\xi$ are mutually independent standard Gaussian variables. In this case, shared features $C\_1=f\_1(X\_1)$ and $C\_2 = f\_2(X\_2)$ cannot achieve exact alignment, i.e., when maximizing mutual information between $C\_1$ and $C\_2$, it is inevitable to include additional redundancy brought by $\xi$ since $C^\*$ can only enter $C\_1$ together with $\xi$.  Now, minimizing $I(Z\_1, C\_1)$ for disentanglement learning in modality 1 will lead to loss of information in $\xi$ in the learned modality-specific feature, while replacing $C\_1$ with $C\_2$ resolves this issue (please see Section 3.2 for more details).
>
> - We have updated all experiments using the revised objective function (and a revised reconstruction loss term detailed below), and notably, with the MOSI dataset, which has a very small sample size–in which case, the CLIP learned features can be undesirable due to a limited sample size, the updated algorithm leads to significant improvement in accuracy (see Table 1).
>
> - In hindsight, the updated objective function is a natural choice. When trying to learn modality-specific features in, say, modality 1, it is possible that $C\_1 = f\_1(X\_1)$ contains some information that is specific to modality 1. In contrast, any information that $C\_2 = f\_2(X\_2)$ contains about modality 1 is, by definition, shared.
> We have already updated experiments using the cross-modal shared features, and notably, with the MOSI dataset, which has a very small sample size–in which case, the CLIP learned features can be undesirable due to a limited sample size, the updated algorithm leads to significant improvement in accuracy (please see Table 1 for more details).
>
>
>
> [**Error bars**] Thank you for pointing this out! We have now added error bars and standard errors for all experimental results.
> Additionally, motivated by the previous question and questions of Reviewer hMRU, we have improved IndiSeek via implementing (1) cross-modal disentanglement objective to address the imperfection of shared features; and (2) a rescaled reconstruction loss to balance the magnitude of two terms. We have updated our draft to implement these changes, and additionally, we caught the mismatch of torchtext versions in previous implementations and have resolved this to ensure correct text data types. The updated experimental results are presented in the newly uploaded version of draft and the improvement of IndiSeek is more pronounced across datasets by incorporating reviewers’ suggestions.
>
> [**Error in density estimation**] Thanks for raising this question! It is indeed true that the estimation error in densities may affect the form of CLUB loss. In synthetic results, even if the distribution of $Z_1=h\_1(X)$ is not Gaussian, as is shown in Figure 1 and Figure 2 panel c, the CLUB loss can still effectively enforce the independence between $C\_1$ and $Z\_1$, which, to some extent, justifies the robustness of the property of the CLUB loss as an upper bound for mutual information.
>
> [**Feature capacity**] We thank the reviewer for pointing out this very insightful question! Actually, in some existing works (e.g., [1]), it has been shown that the performance of CLIP tends to saturate even if the output dimension increases, which is consistent across a range of datasets. We have also added ablation studies in Table 2 with varying output dimensions, from which we can see that the performance of IndiSeek, as well as other baselines, are robust to the output dimension, and across all output dimensions, IndiSeek consistently outperforms in downstream accuracy.
>
> For additional revisions made, please see "Reply to all reviewers" above.

---

### Official Review · Reviewer_hMRU · 2025-11-01

**Soundness:** 2
**Presentation:** 3
**Contribution:** 2
**Rating:** 2
**Confidence:** 4

**Summary:**

The paper proposes a modified, practical criterion for disentangling modality specific features from features shared across the modalities. The main contribution in the paper is the author's criterion for finding modality specific features when shared features have already been established in the previous (typical) step.

**Strengths:**

Many data analysis tasks today are multimodal. Most techniques for disentanglement are variations on a theme. The authors focus on reducing the criterion for modality specific features into a practical form with reasonable motivation. While the conceptual two-step procedure from shared to specific seems similar to other methods, the reduction into a practical approach is different. Section 2 provides a good tutorial introduction for desiderata.

**Weaknesses:**

The use of synthetic examples should be to illustrate generalizable patters in simple settings. The examples shown in the paper appear to have been constructed a bit too intricately without convincing rationales. Regardless of whether authors' method typically dominates others, it is possible to find specific cases where the trends are reversed. Such examples are not convincing. You could instead take the graphical model that all the methods essentially agree with, randomly select standard non-linear dependencies and associated parameters for the conditionals, and show how the methods compare in the resulting disentanglement tasks, including how well they succeed in their respective optimizations.

eq (1) in the paper doesn't seem correct. In the approach of Wang et al., shared and specific encodings are estimated in two steps. The second step for modality specific representation makes use of the already inferred shared representation from the other modality, not the same modality as written in eq (1). Which version did the authors use in comparisons?

Paper isn't self-contained, e.g., NCE-CLUB loss is not given in the form that can be estimated (e.g., using a learned plug-in critic from NCE).

Due to elaborate steps in defining performance in the CITE-seq exmaple, it is hard to use it as a measure of quality among disentanglement methods. Cell annotations are based on weighted clustering (Hao et al) and the authors' theta(c) per cell is constructed by taking 10 nearest Euclidean neighbors in the respective estimated modality specific features and measuring the annotated label consistency among neighbors. It is unclear how the resulting agreement with RNA modality weights (a specific construction used in Hao et al) indicates representational quality.

**Questions:**

How is NCE-CLUB implemented?
Which steps are used to ensure that the methods used for comparison succeed in optimizing their respective criteria?
In the comparisons, are the hyperparameters optimized for authors' method also used for the other methods?

---

> ### Author Response · Authors · 2025-11-24
>
> We thank the reviewer for insightful questions, and we will address the questions and the mentioned weakness as follows. For additional revisions made, please see "Reply to all reviewers" above.
>
> [**Design of synthetic examples**] Thank you for raising this question on the intuition behind synthetic examples, and we would like to add some clarifications. In Example 1, shared and specific features are clearly separated among the coordinates of X, given that the shared map is applied entrywisely, learned modality-specific features as a function of $X$ would be much easier to satisfy both disentanglement and reconstruction; thus, we use Example 1 as a baseline. However, in Example 2, the first four coordinates of X are generated by the 2-dimensional “shared features” in a redundant way; thus, learning specific features $Z_1=h_1(X)$ can be challenging in the sense that, without a good balance of two objectives, Z_1 may contain the cross terms of $X\_1$ and $X\_2$, which will make the disentanglement fail. If desired, we would be glad to further include graphical model demonstrations of the two settings.
>
> [**Cross-modal shared features**] Thanks for this very insightful question! This question is also closely related to “information leakage” mentioned by Reviewer 4mW7.
> - In (Wang et al., 2024), as the reviewer has mentioned, the objective for modality 1 takes the form $\min - I(Z\_1, C\_2; X\_1) + \lambda \cdot I(Z\_1; C\_1)$ which encourages the independence between $Z\_1$ and $C\_1$ while encouraging $(Z\_1,C\_2)$ to reconstruct $X\_1$. However, this could cause issues as follows. Consider a scenario where there is an underlying shared component C^\* between $X\_1$ and $X\_2$, in which case, with information leakage, we can assume $C\_1 = K\_1(C^\*, X\_1)$ and $C\_2 = K\_2(C^\*, X\_2)$. In this case, if we directly enforce the independence between $C\_1$ and $Z\_1$, it will lead to information loss in learned $Z\_1$, and in the meantime, this $Z\_1$ together with C_2 cannot recover X_1, which indicates that if we replace $C\_1$ with $C\_2$ in the reconstruction loss, two terms cannot be minimized simultaneously when there is information leakage in $C\_1$.
>
> - This insightful question indeed motivates the improvement of our procedure, and we have modified the optimization objective to  $\min - I(Z\_1, C\_1; X\_1) + \lambda \cdot I(Z\_1; C\_2)$, i.e., we replace $C\_1$ by $C\_2$ in the disentanglement term to address the scenario with imperfect shared features. We have also updated our draft as well as all our experiments based on the new formulation.
>
> - Consider a simple example where $X\_1 = (C^* + \sin(\xi), Z^*\_1)$ and $X\_2 = (C^\*, Z^\*\_2)$ where $C^\*$, $Z^\*\_1$, $Z^\*\_2$, and $\xi$ are mutually independent standard Gaussian variables. In this case, shared features $C\_1=f\_1(X\_1)$ and $C\_2 = f\_2(X\_2)$ cannot achieve exact alignment, i.e., when maximizing mutual information between $C\_1$ and $C\_2$, it is inevitable to include additional redundancy brought by $\xi$ since $C^\*$ can only enter $C\_1$ together with $\xi$.  Now, minimizing $I(Z\_1, C\_1)$ for disentanglement learning in modality 1 will lead to loss of information in $\xi$ in the learned modality-specific feature, while replacing $C\_1$ with $C\_2$ resolves this issue (Section 3.2).
>
> - We have updated all experiments using the revised objective function, and notably, with the MOSI dataset, which has a very small sample size–in which case, the CLIP learned features can be undesirable due to a limited sample size, the updated algorithm leads to significant improvement in accuracy (see Table 1).

---

> ### Author Response · Authors · 2025-11-24
>
> [**Implementation of CLUB**] We thank the reviewer for pointing this out! We follow the implementation in the original CLUB paper [1]. As CLUB loss is a variational upper bound for mutual information, we use an auxiliary conditional density $q_{\theta}(z | c)$ as an approximation for $p(c | z)$, where $q_{\theta}(z | c)$ lies in the Gaussian family, and we optimize the moments of the Gaussian distribution using 5-layer MLPs in each epoch. The inner optimization is run for 5 epochs at each epoch to optimize representations, and the implementation is the same for both IndiSeek (we use cross-modal shared features) and FactorCL for fair comparison (please see Appendix A.1 for more details).
>
> [**Choice of $\lambda$**] Thanks for the question on the selection of this hyperparameter, which is a crucial ingredient in our approach.
>
> To better understand the tradeoff between two terms and relieve the dependence on feature scales in the “optimal” \lambda in various examples, we consider a scale-free version of the reconstruction loss, in which, instead of $\Vert h(C,Z) - X\Vert^2$, we use $s^{-2}(X)\Vert h(C,Z) - X\Vert^2$ where $s^2(X)=\Vert X\Vert^2$. Then, with this scaled reconstruction loss, we rerun the simulations as well as experiments and present the updated results in the draft. With the rescaled reconstruction loss, the CLUB upper bound and the reconstruction term are now both scale-free, and our experimental results have suggested a universal candidate range of $\lambda$'s between $10^{-1}$ to $10^1$ (as shown in Table 4-8).
>
>
> [**Interpret CITE-seq results**] To interpret the representations learned for CITE-seq, we have added a UMAP for IndiSeek learned representations in Appendix B.3 Figures 18 and 20 as well as those for Factorized CL and InfoDisen in Figures 19, 20, 21, and 22. We can see that the representations exhibit a clear clustering structure at both cell type annotation levels. This is also confirmed by clustering quality metrics such as ARI and NMI: IndiSeek outperformed the two baselines in both scores in both modality-specific and overall learned representations.
>
> ---
>
> [1] Cheng, Pengyu, Weituo Hao, Shuyang Dai, Jiachang Liu, Zhe Gan, and Lawrence Carin. "Club: A contrastive log-ratio upper bound of mutual information." In International conference on machine learning, pp. 1779-1788. PMLR, 2020.

---

### Author Response · Authors · 2025-11-24
**Reply to all reviewers**

We thank all reviewers for their detailed feedback and insightful questions on our paper. These questions have motivated us to revise the manuscript substantially during the rebuttal period, which have significantly improved our work in terms of both conceptual formulation and practical performance. We would like to highlight the major improvements as follows.

While the updated rebuttal procedure has prevented the reviewers from evaluating the major improvements we have made to this manuscript, we hope that the AC could take a close look at the updated manuscript and make a final decision based on its quality after the revision.

---

## Key concerns addressed

**(1) Potential information leakage in shared features and cross-modal switching of shared features**

Reviewers (hMRU, D7zU, 4mW7) have raised questions regarding what representations disentanglement methods would learn when the shared features are either redundant or exhibit information leakage.

- This insightful question indeed motivates the improvement of our procedure, and we have modified the optimization objective in Step 2 to  $\min_{Z\_1} - I(Z\_1, C\_1; X\_1) + \lambda \cdot I(Z\_1; C\_2)$, i.e., we replace $C\_1$ by $C\_2$ in the disentanglement term to address the scenario with imperfect shared features. We have also updated our draft as well as all our experiments based on the new formulation.

- Consider a simple example where $X\_1 = (C^* + \sin(\xi), Z^*\_1)$ and $X\_2 = (C^\*, Z^\*\_2)$ where $C^\*$, $Z^\*\_1$, $Z^\*\_2$, and $\xi$ are mutually independent standard Gaussian variables. In this case, shared features $C\_1=f\_1(X\_1)$ and $C\_2 = f\_2(X\_2)$ cannot achieve exact alignment, i.e., when maximizing mutual information between $C\_1$ and $C\_2$, it is inevitable to include additional redundancy brought by $\xi$ since $C^\*$ can only enter $C\_1$ together with $\xi$.  Now, minimizing $I(Z\_1, C\_1)$ for disentanglement learning in modality 1 will lead to loss of information in $\xi$ in the learned modality-specific feature, while replacing $C\_1$ with $C\_2$ resolves this issue (Section 3.2).

- We have updated all experiments using the revised objective function, and notably, with the MOSI dataset, which has a very small sample size–in which case, the CLIP learned features can be undesirable due to a limited sample size, the updated algorithm leads to significant improvement in accuracy (see Table 1).

**(2) Choice of $\lambda$**

To better understand the tradeoff between two terms and relieve the dependence on feature scales in the “optimal” $\lambda$ in various examples, we have updated the reconstruction loss to a scale-free version in the revised manuscript, in which, instead of $\Vert h(C,Z) - X\Vert^2$, we use $\Vert h(C,Z) - X\Vert^2/\Vert X\Vert^2$. Then, with this revised reconstruction loss, we rerun experiments and present the updated results in the draft. With the rescaled reconstruction loss, the CLUB upper bound and the reconstruction term are now both scale-free, and our experimental results have suggested a universal candidate range of $\lambda$'s between $10^{-1}$ to $10^1$ (Table 4-8).

**(3) Ablation studies and additional experiments**

- [Sensitivity against representation dimensions: D7zU] Ablation studies with varying output dimensions to investigate the sensitivity to feature capacity added (Table 2).
- [CITE-seq without dimension reduction: 4mW7] For CITE-seq dataset, we also present the results with the original 2000-dimensional RNA data in Appendix A.3.1, and the relative performance among the three methods remains the same.
- [CITE-seq evaluation: hMRU] To further understand the representations learned by disentanglement methods, we have also visualized them with CITE-seq dataset via UMAPs and quantified the clustering properties using widely-adopted clustering quality metrics such as ARI and NMI, both of which demonstrate the advantage of IndiSeek (Figures 18-23).
- [Baselines: 4mW7,gHk3] Comparison with an additional baseline CoMM on the MOSI dataset added in Table 10.
- [Error bars: D7zU,4mW7] We have clarified the implementation details of CLUB loss and have included error bars and standard errors at all applicable places.
- [Computational cost: gHk3] We have also compared the computational cost of all three methods and present the results in Appendix B.4.2 (Table 9).

---

## Summary of Revisions

**(a) Implementation of cross-modal switching of shared features**

We implemented the cross-modal switching of shared features to address the potential redundancy or information leakage in shared features in all experiments. A more detailed description of the formulation can be found in the revised Section 3.

**(b) Implementation of rescaled MSE**

Please see **(2) Choice of $\lambda$**

**(c) Visualization and evaluation of CITE-seq representations**

Please see **(3)**[CITE-seq evaluation].


**(d) Additional ablation studies**

Please see the summary of ablation studies above.

---

### Note · Authors · 2026-01-26

I have read and agree with the venue's withdrawal policy on behalf of myself and my co-authors.

---

### Meta-Review · Area_Chair_3qg4 · 2026-01-07

**Summary:**

Reviewers agree that the problem is important and that the paper is generally well motivated, but they raise substantial concerns regarding (i) information leakage and the correctness of the original objective formulation, (ii) clarity and realism of the theoretical assumptions, (iii) evaluation design and interpretability of empirical results, and (iv) the strength and distinctiveness of the contribution relative to prior disentanglement methods.

While the rebuttal meaningfully improves the correctness and clarity, some core concerns—particularly regarding the robustness of disentanglement under imperfect shared features, the interpretability of empirical gains, and the incremental nature of the contribution—remain only partially resolved. More importantly, given that the objective optimization function has been modified and all experiments were reruned, it indicates that the original formulation was incomplete.

**Reviewer Concerns:**

Concerns substantially addressed by the rebuttal include the correctness of the objective and information leakage, additional ablation studies and experiments, the addition of standard errors across all reported experiments, and visualization and evaluation of CITE-seq representations. However, the following concerns are only partially addressed or remain outstanding:

- Information leakage and guarantees of disentanglement: The authors state that avoiding leakage from to is not the goal, but rather isolating the underlying shared information. However, this reframing weakens the original claim: the method relies heavily on the assumption that Step 1 yields a good enough shared representation. Thus, while improved, the concern remains only partially resolved.

- The responses regarding (1) the reliance on linear classifiers for the reported results, (2) diffusion-based decoders, and (3) handling missing modalities at inference are not explicitly found in the rebuttal.

**Reviewer Scores:**

**Reviewer hMRU (Score: 2, reject):**

The rebuttal addresses some technical criticisms, but given this reviewer’s strong skepticism about synthetic examples, evaluation validity, and objective function, a significant score change appears unlikely.

**Reviewer D7zU (Score: 4, marginally below acceptance)**

The rebuttal addresses some of these points, but not enough to clearly push the assessment above the acceptance threshold.

**Reviewer 4mW7 (Score: 4, marginally below acceptance):**

The reviewer acknowledges improvements and additional experiments, but explicitly maintains residual concerns about information leakage, and the nature of representation learning versus decomposition on precomputed embeddings. A slight positive adjustment is possible, but the review remains cautious.

**Reviewer gHk3 (Score: 4, marginally below acceptance):**

This reviewer is generally positive about the framework but some responses are not explicitly found in the rebuttal. As a result, a slight positive adjustment is possible, but the assessment would likely remain in the borderline range.

---

### Decision · Program_Chairs · 2026-01-26

Reject